# End-to-end Planner Training for Language Modeling

## Abstract

Through end-to-end training to predict the next token, LLMs have become valuable tools for various tasks. Enhancing their core training in language modeling can improve numerous downstream applications. A successful approach to enhance language modeling uses a separate planning module to predict abstract labels of future sentences and conditions the LM on these predictions. However, this method is non-differentiable, preventing joint end-to-end tuning of the planner with the LM. We propose an effective method to improve this approach by enabling joint fine-tuning of the planner and the LM. We show that a naive way of approximating the gradient of selecting a label via the straight-through estimator is not effective. Instead, we propose to use the predicted label probabilities as mixing weights to condition the LM on a weighted average of label embeddings in a differentiable manner. This not only enables joint fine-tuning of the planner and the LM, but also allows the LM to draw on the full label distribution predicted by the planner, retaining more information. Our experimental results show consistent improvements in perplexity.

## 1 Introduction

Large Language Models (LLMs) currently lay the foundation for excellent performance across a variety of downstream tasks. They are not trained specifically for these tasks, but are pretrained in next token prediction, on trillions of tokens (typically followed by a short Reinforcement Learning from Human Feedback (RLHF) phase). Improving the effectiveness of the language modeling phase can be expected to lead to greater usefulness on many downstream tasks.

Cornille et al. (2024) are able to improve perplexity and generation quality by 1) pretraining a separate planning module to predict an abstract, discrete label of the next sentence (which they call "writing action", produced through unsupervised clustering) and 2) teaching the LM to condition on the planner's predictions while doing low-level next token prediction. This enables factorizing the language modeling task into prediction of abstract, high-level information processing and concrete, low-level generation conditioned on the high-level plan. Motivated by the idea of keeping the trained planner compatible with any LM, Cornille et al. (2024) do not fine-tune the planner during step (2). However, this means that the whole system is not trained end-to-end, breaking with one of the core concepts that make deep learning, and thus LLMs, so successful.

Therefore, the objective of this paper is to further improve LM performance by enabling end-to-end joint fine-tuning of the planner and the LM. This comes with two challenges: **(i)** The discrete choice of an action is non-differentiable. **(ii)** Useful high-level features that the planner has learned during step (1) may be forgotten due to the low-level objective in step (2).

In order to address challenge **(i)**, we first investigate the straight-through estimator as a method for approximating the gradient of the hard-selection of a writing action. Finding that this is not effective, we propose a method that uses the planner-predicted action probabilities to compute a weighted average of the action embeddings, rather than selecting an embedding based on which action has the highest probability. This has two advantages: First, it allows us to compute an exact gradient. Second, it does not discard valuable information contained in the label distribution predicted by the planner. We perform probing experiments that corroborate the hypothesis that using the full range of planner-predicted probabilities results in representations that are more informative about distant tokens.

To address challenge **(ii)**, we show two approaches that succeed at preventing catastrophic forgetting: one approach is to keep the planner parameters frozen for the first half of training before unfreezing them for the second half of training, another approach is to fine-tune the planner with a mix of its original Next-Action Prediction objective and the Next-Token Prediction objective. Moreover, we show that pretraining the planner with only the low-level objective (Next Token Prediction) damages performance, again underscoring the importance of the high-level objective (Next Action Prediction).

As observed in prior work, generation metrics can be improved by training the language model with oracle actions, as this means the language model learns to rely on the plan more strongly. However, this teacher forcing damages perplexity due to the mismatch of oracle actions during training with imperfect planner-predicted actions during evaluation, known as exposure bias. We show that we can strike a balance in this trade-off by mixing oracle and planner-predicted actions with a certain probability during training.

Our contributions can be summarized as follows:

- We propose a method that addresses the challenges of joint end-to-end training of a high-level planner module and a low-level language model.

- We compare our method using two LM backbones (GPT-2-small and OLMo-1B), demonstrating improvements in perplexity.

- We show that improvements depend on two key factors: (i) avoiding catastrophic forgetting of the planner's high-level knowledge—achieved by delaying the unfreezing of planner parameters or including the planner's high-level objective during training; and (ii) ensuring the LM has access to all planner-predicted probabilities rather than just the top prediction.

- We identify a trade-off between perplexity and generation quality that depends on the use of oracle or planner-predicted actions during training, and demonstrate that mixing oracle and planner-predicted actions balances this trade-off effectively.

## 2 Related Work

Our method is related to four fields of research. 1) Conditioning the language model on a writing action is akin to *Controllable Text Generation*, where a language model is conditioned on attributes of various types, such as content or style, to guide generation. 2) Since the planner operates at a higher level than the LM, our method is related to *Hierarchical Language Modeling*. 3) As the core contribution of our paper is enabling end-to-end training, we discuss techniques for *bridging the differentiability gap*.

**Conditioning language models on plans** In Controllable Text Generation (Zhang et al., 2024), human-provided inputs are used to condition text generation, but we use latent, planner-produced inputs rather than human-provided ones. Chain-of-thought reasoning (Wei et al., 2023) improves outputs through explicit, token-level intermediate steps, and is not learned explicitly but only as a byproduct of LM pretraining. In contrast, the approach we build on conditions outputs on compact, latent-space plans, where a single plan embedding covers a sentence rather than individual tokens, and explicitly trains the model to condition generation on these plans. Quiet-STaR (Zelikman et al., 2024) conditions next token prediction on a model-produced input, but this is applied at every token, making it more computationally intensive, and is in text space rather than latent space. Wang et al. (2023) use planning tokens with a similar objective as the writing actions we consider, but they use an *internal* planner, i.e., the LM itself, to predict planning tokens, in contrast to the external planner from Cornille et al. (2024), who found an internal planner to be less flexible and less effective.

**Hierarchical Language Modeling** Chung et al. (2017), Li et al. (2022), and Subramanian et al. (2020) all factorize language modeling into a per-token part and a slower 'per text-unit' part (e.g., per-phrase, per-sentence, or every fixed number of tokens). The per text-unit component in their work is only optimized for predicting concrete tokens. The key difference with this work is that our per text-unit component (i.e., the planner) is pretrained to predict targets in an abstract sentence space, and only then fine-tuned together

with the per-token component (i.e., the LM) to predict concrete tokens. Marfurt & Henderson (2021) and Jhamtani & Berg-Kirkpatrick (2020) also aim to improve language modeling by decomposing at the sentence and word level, but unlike Cornille et al. (2024) and this work do not operate at an *abstract* sentence level.

**Bridging differentiability gap** A key component of our contribution is to enable the planner to be optimized jointly with the LM. A number of methods exist aimed at estimating a gradient for non-differentiable operations such as an argmax. Policy Gradient methods are one type of approach which are common in reinforcement learning (Williams, 1992; Sutton et al., 1999; Greensmith et al., 2004; Schulman et al., 2017). While unbiased, these tend to have high variance. Relaxed gradient estimators (Maddison et al., 2016; Paulus et al., 2020) constitute another approach, they replace a discrete sample with a continuous variable to calculate the gradient. An important variant of these is the straight-through estimator (Bengio et al., 2013), which uses a hard max for the forward pass, but a softmax for the backward pass. While this is a popular biased, but low-variance estimator of the gradient, we demonstrate that it yields unsatisfactory results in our scenario.

**Exposure bias and teacher forcing** The method proposed by Cornille et al. (2024) learns a *cascaded model*, where the predicted class of the first model A (the planner) is propagated into the second model B (the language model). An old technique in such cases is to replace the actual prediction of model A with the ground truth signal (Jordan, 1986; Pineda, 1988). In sequence learning networks such as RNN this is known as *teacher forcing*, which can help stabilize training (Williams & Zipser, 1989). However, on the flip side, this approach can introduce an *exposure bias*, in which the distribution of inputs seen by model B at training time differs from inference time, leading to sub-optimal results (Bengio et al., 2015; Lamb et al., 2016). There are several ways to circumvent the exposure bias problem. The first type of approach aims to mitigate the training and test distribution mismatch. Scheduled sampling (Bengio et al., 2015) selects the self-predicted input with probability $p$ and the ground truth input with probability $1 - p$ during training. $p$ is scheduled to increase from 0 to 1 over the course of training to retain the best from both worlds. Professor Teaching (Lamb et al., 2016) uses an adversarial method to make the train and test distribution similar. Another type of approach circumvents the problem altogether by training the model end-to-end on the metric of interest, e.g. Ranzato et al. (2015) use self-predicted outputs in combination with reinforcement learning for text generation, and Graves & Jaitly (2014) directly optimizes for low word error rate in speech recognition. In this paper, we explore both scheduled sampling and end-to-end training to deal with the exposure bias.

## 3 Methodology

We summarize the common elements we preserve in 3.1. We then detail the prior way of interfacing the planner and the language model in 3.2, and our novel way in 3.3. We illustrate common elements and the contrast between the prior and novel component in Figure 1.

### 3.1 Overview of Common Methodology

We consider the task of language modeling, where the goal is to estimate the probability $p(x_1 x_2 \ldots x_n)$ for any text sequence $X = x_1 \ldots x_n \in \mathcal{X}$. We also refer to this task as *Next Token Prediction*. The probability is factorized into the product of probabilities of each token given the preceding tokens: $p(x_1 \ldots x_n) = \prod_{i=1}^{n} p(x_i | x_1 \ldots x_{i-1})$. Unlike a standard LM, the predicted probability in our method is conditioned on additional predictions by an external planner.

To obtain training data for this planner, we apply a pretrained text encoder (e.g., Sentence-BERT) to each sentence $t_j$ in the corpus to produce a corresponding low-dimensional vector $\mathbf{z}_j$. K-means clustering on all embedded text units from the corpus yields clusters that are used as abstract labels (or 'writing actions') $a \in \mathcal{A}$. A cluster's centroid serves as the (initial) action embedding. This is shown in step 1 in Figure 1.

The planner module is then pretrained to predict the writing action $a_i$ that corresponds to sentence $t_i$ based on the context of the preceding text units $t_1, \ldots, t_{i-1}$. This task is called *Next Action Prediction*. The planner module is based on a custom Transformer architecture that first embeds each sentence independently

into a single vector and then contextualizes them with a Transformer encoder. This is shown in step 2 in Figure 1. For more details, please refer to Cornille et al. (2024).

## 3.2 Prior approach to Language Model Fine-tuning

In Cornille et al. (2024), the input $X$ is split into $m$ sentences $X = t_1 \ldots t_m$, where each sentence $t_j = x_1^j \ldots x_{n_j}^j$ has a single associated *predicted* writing action $\hat{a}_j$, predicted by a pretrained planner. The fine-tuning objective then is to estimate $\prod_{j=1}^m \prod_{i=1}^{n_j} p(x_i^j | \hat{a}_1 \ldots \hat{a}_j, t_1 \ldots t_{j-1} x_1^j \ldots x_{i-1}^j)$.

The predicted actions are one-hot encoded, and used as index to select action embeddings in the action embedding matrices $E_A^l$ that are added at various layers $l$ of the LM, resulting in a vector $r_j^l = E_A^l(\hat{a}_j)$. That vector is then projected and element-wise added to each hidden representation in layer $l$, just after the multiplication with attention weights and just before the output projection, similar to Llama-Adapter (Zhang et al., 2023).

Importantly, planner parameters are not updated during LM fine-tuning, disallowing the planner to improve its action predictions beyond matching cluster labels, by tailoring them to the LM.

## 3.3 Novel Joint Planner-Language Model Fine-tuning

We hypothesize that the planner can enhance the utility of its predictions by being fine-tuned for next-token prediction, jointly with the LM, after being pretrained for Next Action Prediction. Hence, we want to enable the gradient to be passed into the planner.

A naive way to achieve this is using a straight-through estimator, which involves using a hardmax function for the forward pass but a softmax function for the backward pass during gradient computation:

$$\text{hardmax}(\mathbf{s}) = \text{onehot}(\text{argmax}(\mathbf{s})) \tag{1}$$

$$\text{softmax}(\mathbf{s}) = \left[ \frac{e^{s_1}}{\sum_{j=1}^{|\mathcal{A}|} e^{s_j}}, \frac{e^{s_2}}{\sum_{j=1}^{|\mathcal{A}|} e^{s_j}}, \ldots, \frac{e^{s_{|\mathcal{A}|}}}{\sum_{j=1}^{|\mathcal{A}|} e^{s_j}} \right] \tag{2}$$

Here, $\mathbf{s} = [s_1, \ldots, s_{|\mathcal{A}|}]$ is the vector of planner-predicted logits for each of $|\mathcal{A}|$ possible actions, and $\text{onehot}(i)$ returns a vector of length $|\mathcal{A}|$ where the $i$-th element is 1 and the rest are 0 (one-hot encoding).

In the forward pass, we compute the action selection using the hardmax function (Eq. 1), which produces a one-hot encoded vector indicating the action with the highest logit. However, during backpropagation, we use the softmax function (Eq. 2) to approximate the gradients, effectively allowing gradients to flow through the non-differentiable argmax operation.

The straight-through estimator is limited by being a biased estimator of the gradient however. A more effective method arises from the insight that there is no inherent need to select only one action. Instead, we can use the planner probabilities to obtain a weighted average of the action embedding columns:

$$\mathbf{r}_j^l = \sum_{a \in \mathcal{A}} \text{softmax}(\mathbf{s})_a \cdot E_A^l(a), \tag{3}$$

Step 3 in Figure 1 illustrates this approach.

This approach not only provides an exact gradient but also allows the LM to make use of the full set of probabilities assigned by the planner to each writing action, rather than only the most probable action.

**Preventing catastrophic forgetting** Unfreezing the planner immediately along with the LM poses the risk of catastrophic forgetting (French, 1999) of the learnings from the Next-Action Prediction pretraining stage. One common mitigation strategy is to unfreeze only after some training steps (Howard & Ruder,

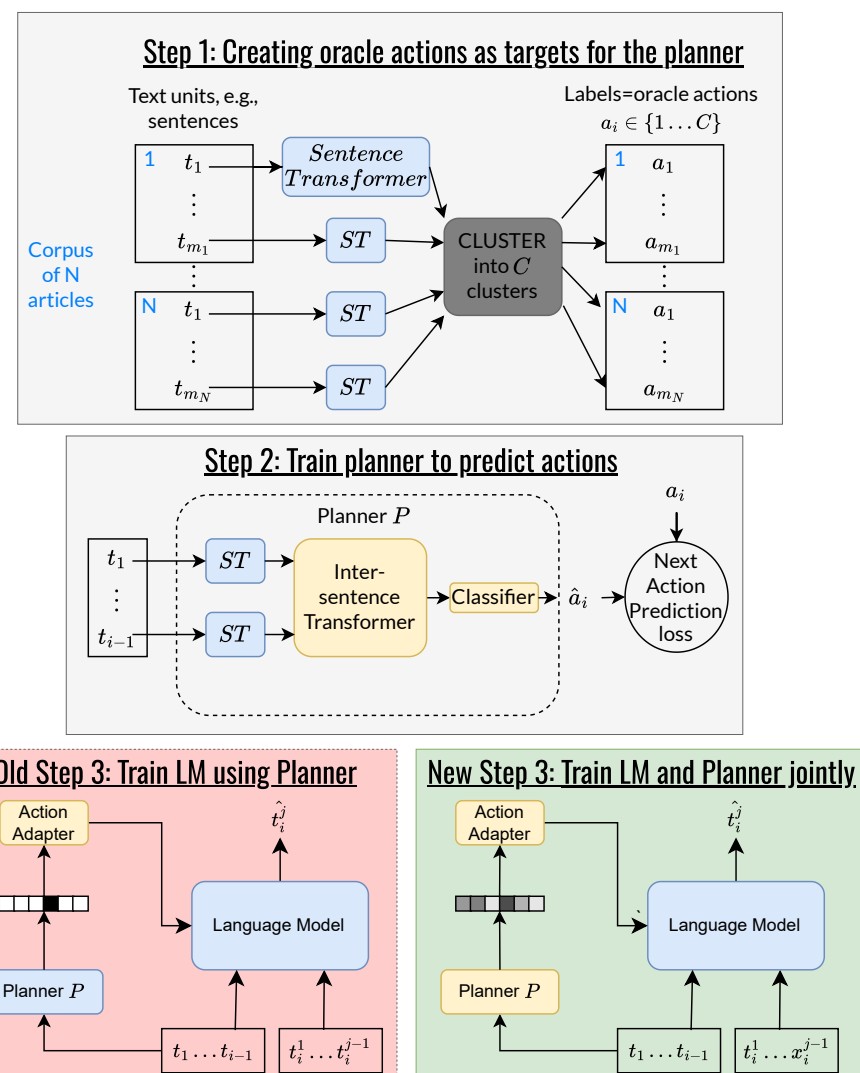

Figure 1: Illustration our proposed improvement to the planner proposed in Cornille et al. (2024). Step 1 and Step 2 are the same as in Cornille et al. (2024). In Step 3, instead of using a single predicted action, the planner predicts a distribution over actions, which is used as mixing weights to compute a weighted average of the action embeddings. This allows the planner to be fine-tuned jointly with the LM. Blue indicates frozen parameters, yellow indicates trainable parameters, and grey indicates no learnable parameters.

2018), hence we evaluate a setting in which we unfreeze the planner only halfway through training.[1] Another strategy we evaluate is to fine-tune the planner with a mix of its original Next-Action Prediction objective, and the Next-Token Prediction objective.

---

[1]Unfreezing the planner halfway through a single epoch with $N$ data points is analogous to training for two epochs on $N/2$ data points and unfreezing after the first epoch. We opt for the former approach because our constraint is computational budget, not data availability. Given a fixed computational budget, training for one epoch on all $N$ data points allows the model to experience the entire data distribution, which can be more beneficial than training for additional epochs on a smaller subset of the data.

## 4 Experimental Setup

The purpose of our experiments is twofold. First, we want to test our hypothesis that end-to-end joint planner-LM training is beneficial for language modeling performance. Second, we want to validate the design decisions we made to enable end-to-end training: a) Using a soft-selection via weighted average rather than a hard selection, and b) mitigating catastrophic forgetting by unfreezing the planner only after half the training or using a mixed Next Action / Next Token prediction objective.

### 4.1 Dataset and backbone models

We train and evaluate our models on the same dataset as Cornille et al. (2024), i.e., subsets of English Wikipedia articles from the "20220301.en" version from Huggingface[2]. Wikipedia articles have the advantage that they cover an extensive range of topics, while also being structured in a way that makes them well-suited for leveraging an abstract planner. We perform experiments both using the small GPT2 model (Radford et al., 2019) and OLMo 1B (Groeneveld et al., 2024) as starting points for the LM. Model details and hyperparameters are provided in Appendix A.

### 4.2 Evaluation

**Primary evaluation**   Our main metric is perplexity, which is the default metric used for language modeling, corresponding to the inverse geometric mean of the probability of true texts according to the language model.

**Generation evaluation**   As in Cornille et al. (2024), we complement perplexity, which does not directly assess generated text, with generation metrics. We report ROUGE-2 (F1) (Lin, 2004) and MAUVE (Pillutla et al., 2021) to evaluate generated texts at the surface level, and Levenshtein distance (Levenshtein et al., 1966) and latent perplexity (Deng et al., 2022) to assess text quality at an abstract level. For the surface level, ROUGE-2 evaluates bigram overlap between generated and real text, while MAUVE measures the divergence between model and true data distributions by comparing generated and real texts unconditionally. For the abstract level, we first map true and generated texts onto the sequence of writing actions that correspond to them. Levenshtein distance then measures the edit distance between generated and ground-truth writing action sequences, and latent perplexity estimates how well the generated sequence aligns with a latent HMM-based critic trained on real texts. We refer to appendix B for more details about the generation evaluation.

**Probing**   In order to understand how the different training setups influence what information the model (un)learns, we use probing classifiers on top of the (frozen) representations to determine how well they predict the upcoming tokens. The choice of probing classifier is not straight-forward (Belinkov, 2022). We choose linear probing classifiers to measure to what extent the information about upcoming tokens can be easily extracted (i.e., is linearly separable) from the representations, rather than be learned by the probe itself (Alain, 2016).

We probe representations at two kinds of locations inside the model. First, we probe the output from the action embedding inside the adapter, which contains information only from the planner (Pre-merge). Second, we probe the representation after the information of the planner has been mixed with the contextualized information from the LM itself (Post-merge). We train probes at every layer where the planner information is infused.

Figure 2 illustrates the two kinds of probing locations.

### 4.3 Settings

**Variations of end-to-end planner**   We evaluate the impact of 4 properties of the end-to-end planner.

---

6

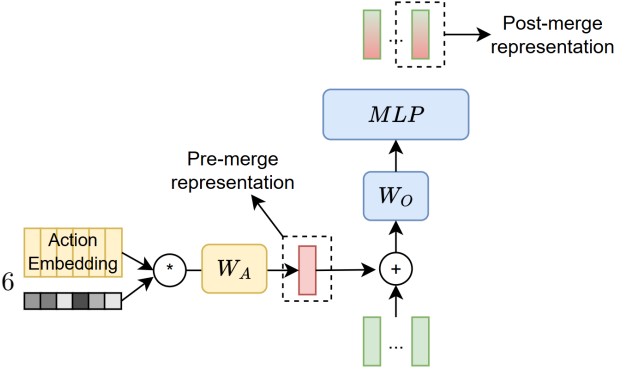

First, whether the planner's parameters are unfrozen immediately (*Unfrz immediate*), halfway through training (*Unfrz halfway*), or never (*Unfrz never*). We expect that immediately unfreezing the planner when the LM hasn't adapted to it yet might lead to catastrophic forgetting, while not unfreezing it at all doesn't allow the planner to tune itself to the LM.

Second, similarly aimed at preventing catastrophic forgetting, we evaluate the effect of continuing to train the planner for its Next-Action Prediction objective at the same time as also tuning end-to-end for Next-Token Prediction Objective.

Third, we evaluate the effect of replacing soft-selection via weighted averages (Eq. 3) with hard-selection via the straight-through estimator (Eq. 1 and 3).

Finally, because we are now able to train the planner end-to-end, we evaluate whether its Next-Action-Prediction (NAP) pretraining objective is still necessary by assessing models in which the planner is only pretrained with a Next-Token-Prediction objective. Specifically, we replace NAP training of the planner with an end-to-end stage in which we keep the LM parameters frozen.

**Baselines**  As baseline without any planner, we follow Cornille et al. (2024) in evaluating the model that includes the same additional adapter parameters that are fine-tuned, but always selects the same fixed action embedding, rather than relying on a planner to select action embeddings (*Fixed*).

To rule out that the benefit of soft-selection is not merely due to mixing multiple actions, we train a variant of the soft-selection method that always applies uniform weighting across all actions (*Uniform*).

Our main baselines are the planner models proposed in Cornille et al. (2024). They have two variants: one pretrained on oracle actions (OA), and one pretrained on predicted actions (PA). Cornille et al. (2024) observed a trade-off between these variants: while PA had better perplexity, OA performed better in some generation metrics. We explore this trade-off more in-depth by making models that mix oracle-action and predicted actions during fine-tuning of the language model.

## 5 Results and Discussion

### 5.1 Main results

Table 1 shows our main results.

**Benefit of end-to-end training**  Comparing **Ours (soft-selection)** to the **Baselines**, the results confirm our hypothesis that end-to-end joint planner-LM training can improve language modeling performance compared to the prior approach, with our best setting improving by 0.3 (GPT-2) and 0.08 perplexity (OLMo), respectively over *Cornille et al. (2024) PA* (Predicted Actions).

We observe that this perplexity improvement does not always translate into improved generation metrics. As noted in Cornille et al. (2024), there is a trade-off between perplexity and performance on generation metrics stemming from the use of teacher forcing for the actions. We examine this trade-off in more detail in section 5.3.

| Base LM | GPT2 | | | | | OLMO | | | | |
|---|---|---|---|---|---|---|---|---|---|---|
| Setting | PPL ↓ | MAUVE ↑ | Latent PPL ↓ | ROUGE-2 ↑ | Edit ↓ | PPL ↓ | MAUVE ↑ | Latent PPL ↓ | ROUGE-2 ↑ | Edit ↓ |
| **Baselines** | | | | | | | | | | |
| Cornille et al. (2024) OA | 26.94 | 0.447 | 91.60 | 0.0193 | 3.69 | 11.99 | 0.411 | 76.20 | 0.0278 | 3.26 |
| Cornille et al. (2024) PA | 25.55 | 0.435 | 205.90 | 0.0169 | 3.78 | 11.46 | 0.563 | 178.20 | 0.0253 | 3.31 |
| Fixed | 26.69 | 0.379 | 352.80 | 0.0154 | 3.88 | 11.81 | 0.445 | 250.90 | 0.0217 | 3.42 |
| Uniform | 26.69 | 0.378 | 354.27 | 0.0159 | 3.91 | 11.81 | 0.396 | 256.13 | 0.0219 | 3.43 |
| **Ours (soft-selection)** | | | | | | | | | | |
| Unfrz immediate | 25.42 | 0.423 | 245.48 | 0.0178 | 3.68 | 11.42 | 0.564 | 187.61 | 0.0271 | 3.33 |
| Unfrz halfway | 25.23 | 0.422 | 205.14 | 0.0183 | 3.80 | 11.37 | 0.551 | 163.78 | 0.0270 | 3.23 |
| Unfrz never | 25.32 | 0.420 | 187.54 | 0.0175 | 3.74 | 11.49 | 0.546 | 163.81 | 0.0271 | 3.29 |
| **Ours (straight-through)** | | | | | | | | | | |
| Unfrz immediate | 25.94 | 0.401 | 281.34 | 0.0162 | 3.87 | 11.53 | 0.548 | 208.12 | 0.0229 | 3.36 |
| Unfrz halfway | 25.66 | 0.464 | 230.00 | 0.0171 | 3.79 | 11.42 | 0.547 | 185.76 | 0.0254 | 3.34 |
| **Ours (NAP during fine-tuning)** | | | | | | | | | | |
| Unfrz immediate | 25.24 | 0.459 | 177.34 | 0.0172 | 3.72 | 11.42 | 0.576 | 155.38 | 0.0266 | 3.29 |
| **Ours (no NAP pretraining)** | | | | | | | | | | |
| Unfrz immediate | 25.80 | 0.443 | 271.71 | 0.0167 | 3.83 | 11.69 | 0.562 | 227.46 | 0.0231 | 3.44 |
| Unfrz halfway | 25.82 | 0.397 | 299.62 | 0.0165 | 3.78 | 11.66 | 0.534 | 224.99 | 0.0218 | 3.37 |
| Unfrz never | 26.15 | 0.435 | 299.51 | 0.0163 | 3.78 | 11.80 | 0.403 | 258.24 | 0.0204 | 3.44 |

Table 1: Perplexity and generation metrics under different training and conditioning scenarios. Cells shaded in red show perplexity, those in blue show the generation metrics. A darker color indicates a worse result.

Using soft-selection and end-to-end training does introduce some additional latency, whch we discuss in Appendix F

**Soft selection beats hard selection**  Comparing **Ours (soft-selection)** to **Ours (straight-through)**, we see that soft-selection variants are consistently better than straight-through variants. This can be explained by two factors. First, the biased gradient estimates of the straight-through estimator might lead the learning astray. Second, soft-selection has the benefit of allowing the LM to draw on the full label distribution: In fact, the soft-selection *Unfrz never* result shows that this alone is already beneficial, even without updating the planner. This explanation is corroborated by the probing results presented in section 5.2, which show that linear probes trained on the soft-selected planner output (rather than the hard-selected one) are better able to predict distant tokens.

Soft-selection also activates the full action embedding matrix at every prediction. However, the fact that *Uniform* performs considerably worse shows that just using the full embedding matrix is *not* responsible for the improvement.

**Timing matters for planner unfreezing**  Keeping the planner frozen during part of the training is more effective than either immediately unfreezing the planner or keeping it frozen the entire time. This is in line with our hypothesis that immediately unfreezing the planner leads to big initial gradients that erase some of the useful knowledge built up during the Next-Action-Prediction planner pretraining phase. On the other hand, not unfreezing the planner at all prevents the planner parameters from specializing toward perplexity minimization.

Our alternative approach to preventing catastrophic forgetting (**Ours (NAP during fine-tuning)**) achieves performance nearly on par with unfreezing the planner halfway.

**Next-Action-Prediction objective cannot be left out completely**  Finally, we see that the models we run with **no NAP pretrainnig** are generally worse for both perplexity and generation metrics than **Ours (soft-selection)**. This indicates that the abstract pretraining objective of the planner is still required, even when end-to-end training is possible.

## 5.2   What do the models learn?

Figure 3a shows the results of our probing experiments by distance to the target token. Unsurprisingly, tokens farther away tend to be more difficult to predict. Regarding pre-merge representations, the Cornille et al. (2024) baseline is notably worse than our proposed methods using soft-selected representations, which benefit from making better use of the full planner's predicted scores rather than only the argmax. Generally, the post-merge representations are significantly better than the pre-merge representations. In fact, the language model alone, without any (useful) planner information ("Uniform") already predicts farther into the future than just the next token. However, adding the pre-merge representations of the planner yields further improvements. Moreover, freezing the pretrained planner at least for half the training epoch tends to retain more information about the upcoming tokens than unfreezing it immediately.

While this probing experiment cannot prove a causal mechanism, it is plausible that the improved performance observed in Table 1 is at least partially attributable to the models ability to being better at predicting several tokens ahead.

Figure 3b shows that the information contained in pre-merge representations is largely independent of the layer, which is explained by the fact that lower layer representations do not feed into higher layer representations. In contrast, post-merge results clearly show that higher layers, which are located closer to the output layer that performs the final token prediction, contain more information about the upcoming tokens.

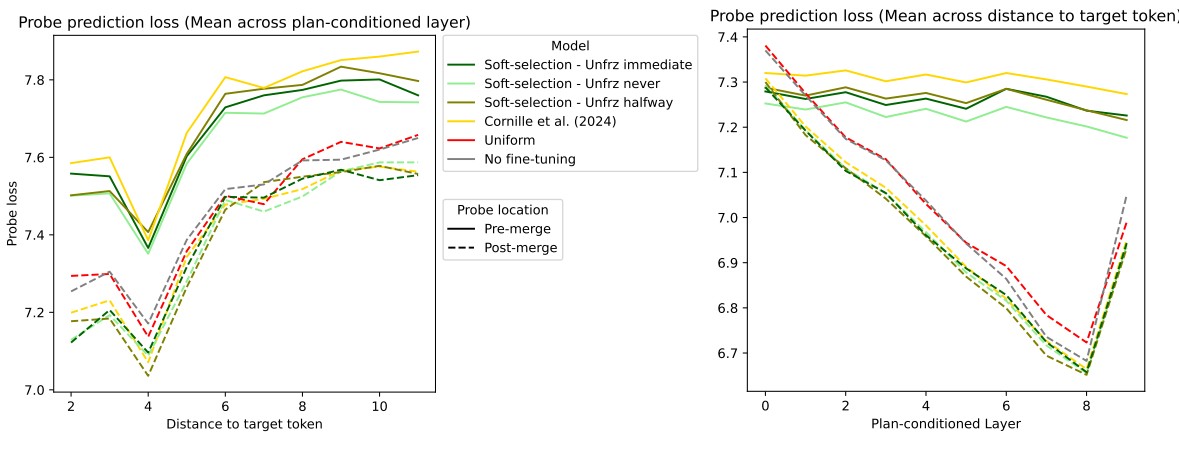

(a) Probe Layer Mean, for tokens at least 2 positions into the future.

(b) Probe Target Mean

Figure 3: Plots showing probing performance at different layers and for different distances to the probe's target token. The distance to target token indicates how many positions into the future the target token is relative to the token immediately following the Language Model input.

## 5.3   Trade-off between Perplexity and Generation Metrics

To investigate the trade-off between perplexity and generation metrics (MAUVE, ROUGE-2, Edit distance and Latent Perplexity), we train models that use a mixture of oracle and planner-predicted actions during training, where we vary the fraction of planner-predicted actions from 0 (equivalent to Cornille et al. (2024) OA) to 1 (equivalent to Cornille et al. (2024) PA). The left side of Figure 4 shows that the smaller the fraction of oracle actions during training, the better the perplexity, up to an improvement of around 5%. Because perplexity evaluation happens with planner-predicted actions, the bigger the fraction of oracle actions during training, the bigger the training/evaluation mismatch, a problem known as exposure bias.

The perplexity improvement does not translate into improving generation metrics, with some metrics (Latent Perplexity and ROUGE-2) even consistently worsening. To understand this, consider the plan-matching accuracy (green line). As fewer actions are oracle, the plan-matching accuracy decreases, indicating the

language model learns to rely less on the plan. This suggests that these generation metrics benefit from having a model rely more on the planner output, even if it is imperfect.

To try to get the best of both worlds, we also evaluate a setting with a scheduled fraction that linearly increases the fraction of planner-predicted actions from 0 to 1 during training, i.e., scheduled sampling, shown on the right side of Figure 4. However, we observe that this leads similar results as training only on planner-predicted actions.

Hence, overcoming this trade-off by both overcoming the problem of exposure bias and ensuring the language model learns to sufficiently rely on the proposed plans is an interesting avenue for future work.

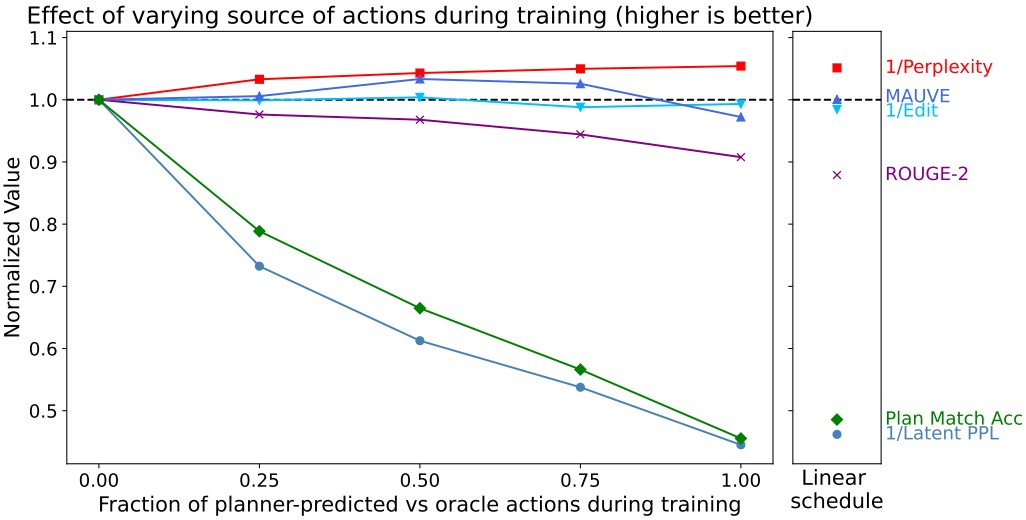

Figure 4: Relative improvement/worsening of our metrics as we increase the fraction of planner-predicted actions from zero (equivalent to Cornille et al. (2024) OA) to one (equivalent to Cornille et al. (2024) PA). Some metrics are inverted, so that higher is better for all metrics.

In Appendix E, we show the same model settings as in Figure 4, but show separate metrics for test examples in which the correct writing action was predicted, and examples in which it was not. As expected, we find that test examples with correct actions score much better.

## 6 Conclusion

Since end-to-end training is a key ingredient to the success of deep learning, it is important that we enable different system components to be optimized jointly. In this work, we bridge the differentiability gap of a recent pretrained planning module with a language model by turning the indifferentiable hard-selection into a differentiable soft-selection. Our results demonstrate that this consistently improves perplexity. We hope these findings can provide a foundation for enhancing production-scale language models through end-to-end planning mechanisms.

# 7 Limitations

## 7.1 Model Size

Due to computational constraints, our evaluation was performed on relatively small models. Consequently, the scalability and effectiveness of the proposed method need to be validated on production-scale models to ensure its applicability in real-world scenarios.

## 7.2 Planning Horizon

Our approach involves planning only one step into the future. This is a simplification compared to how humans presumably think and plan farther into the future. Future work should investigate methods to extend the planning horizon, allowing the model to consider multiple future steps and thereby improve decision-making processes.

### Broader Impact Statement

While increasingly more capable LLMs are very useful, they can also be misused for harmful purposes (such as generating disinformation, helping in development of weapons, etc.). Because our work has used LLMs of modest size, there is little risk of it contributing to such misuses directly. It could do so however if our method would be used to make production-scale language models even more effective. If that is the case, it is important to take the necessary precautions before deployment, such as proper alignment with human values.

The compute requirements of large models also have a significant environmental impact (Rillig et al., 2023). Use of a planning module also entails additional compute requirements, which can further contribute to this, although the planning module is relatively lightweight compared the the Language Models, and is invoked only once per sentence rather than for every token.

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

## A  Model Details

**Parameter counts**  Table 2 shows parameter counts for our models

| Model | Parameter Count |
|---|---:|
| GPT2-Small | 124,439,808 |
| Olmo | 1,176,764,416 |
| Extra conditioning parameters | 13,770,240 |
| Planner parameters | 116,378,496 |

Table 2: Parameter counts for our models

**Computational Budget**  We ran our experiments on either 12GB, 16GB or 24GB GPUs, each time using one GPU per experiment. We report a single run for each setting. With this setting, joint fine-tuning of planner and LM takes around 40 hours for GPT2-Small and 60 hours for Olmo 1B. Pretraining the planner for Next Action Prediction took around 90 hours, but we reuse the same pretrained planner for most experiments. Evaluating perplexity takes about 3-5 hours, while evaluating edit-distance (which requires generation) takes around 10-15 hours.

We estimate that we ran about 100 experiments (only a subset of which led to results presented in the paper), which means in total we used around 7000 GPU hours.

**Used artifacts**  We build on the source code of Cornille et al. (2024), which was shared with us privately. We will release our extensions publicly once their code is made available.

We use spaCy (Honnibal et al., 2020) to split articles into sentences. These sentences are then transformed into embeddings using MPNet-base-v2 (Song et al., 2020) through the SentenceTransformer library (Reimers

& Gurevych, 2019)[3]. The final clustering is conducted using k-means++ initialization (Arthur & Vassilvitskii, 2007) implemented in Scikit-Learn (Pedregosa et al., 2011).

The Wikipedia dataset is accessed via the 'datasets' library at `https://huggingface.co/datasets/wikipedia`, specifically the March 2022 version ('20220301').

We use PyTorch (Paszke et al., 2019), the Huggingface 'datasets' (Lhoest et al., 2021), and 'transformers' (Wolf et al., 2020) libraries for loading and preprocessing data and pretrained models (specifically GPT-2 (Radford et al., 2019)). Additionally, we employ PyTorch-Lightning (Falcon et al., 2020) for model training. All the libraries utilized are open source or freely available for academic use.

**Hyperparameters** Table 3 shows hyperparameters used for our experiments. We do not perform hyper-parameter search for these, using the default hyperparameters reported in Cornille et al. (2024). We use the Adam optimizer (Kingma, 2014), and always train for one epoch.

Table 3: Hyperparameter Settings

| Hyperparameter | Value |
|---|---|
| Context window size | 128 |
| Train \| test \| val split sizes | 285310 \| 1000 \| 1000 |
| K-means initialization | k-means++ |
| Default action count | 1024 |
| Action embedding dimension | 768 |
| **Language Model Fine-tuning** | |
| Batch size | 32 |
| Learning rate | 1e-4 |
| **Planner Training** | |
| Batch size | 32 |
| Learning rate | 1e-4 |

## B  Generation Evaluation Setup and Detailed Results

We follow the evaluation setup from Cornille et al. (2024), and explain the details again here:

For MAUVE and Latent Perplexity, we generate 1024 tokens unconditionally (i.e., without context), matching the average length of the articles in the dataset.

For ROUGE-2, and Edit distance, we use a prefix $t_1 \dots t_i$ from real texts and generate continuations from that prefix of 128, 256, 512, and 1024 tokens. Because Edit distance scales linearly with the number of tokens, we normalize the results across different lengths. For 128 tokens, we report the raw edit distance; for 256, we divide the edit distance by 2, and so on, ensuring a consistent comparison across generation lengths.

The results in the main text (Table 1) are the average for these different generation lengths.

## C  Repeat experiments with different random seeds

Figure 5 compares performance for multiple random seeds. Three models are compared:

- Fixed: a baseline that always receives the same, uninformative, plan

---

[3]`https://huggingface.co/sentence-transformers/all-mpnet-base-v2`

- Hard/Freeze: the model that achieved the best perplexity from Cornille et al. (2024), which uses a planner that selects a single writing action, and doesn't fine-tune the planner parameters towards next token prediction

- Soft/Unfreeze Halfway: the best model from this paper, which uses a mix of writing actions based on planner-predicted probabilities, and unfreezes the planner after some time to be fine-tuned for next token prediction.

It shows that in perplexity, Hard/Freeze consistently improves over Fixed, and that Soft/Unfreeze Halfway consistently improves over Hard/Freeze. For other metrics, models with a planner (Hard/Freeze and Soft/Unfreeze Halfway) are consistently better than Fixed. However, Soft/Unfreeze Halfway does not consistently improve generation metrics compared to Hard/Freeze. We hypothesize this is connected with the trade-off due to plan-matching discussed in section 5.3. Ensuring that the end-to-end planner perplexity benefits translate into improved generation metrics too is an import avenue for future work.

## D  Performance on OpenWebText

For our main experiments, we evaluate on Wikipedia articles because they are structured and likely to benefit from a high-level planner.

In this appendix, we show results for an additional dataset, namely OpenWebText (Gokaslan et al., 2019), an open-source replication of the WebText dataset from OpenAI, that was used to train GPT-2. It covers a broader range of data sources than only Wikipedia articles.

The results are shown in Figure 6.

We observe that the planner proposed by Cornille et al. (2024) does not consistently improve over the baseline for the OpenWebText dataset, while our improved planner does. The improvement over the Fixed baseline is significantly smaller however. We hypothesize that this is because this dataset is more varied and less structured than Wikipedia articles, requiring a more powerful planner with a larger action space trained on significantly more data. We leave this for future work.

## E  Effect of correct versus incorrect action predictions for varying mix of oracle and predicted codes

Figure 7 shows separate metrics for test examples in which the correct writing action was predicted, and examples in which it was not, for setting that differ in their mixture of oracle and planner-predicted actions during training.

Note that for MAUVE and Latent Perplexity, there is no oracle action, because these metrics don't compare individual generations to matching true texts, but groups of generated texts to groups of true texts. Further, for ROUGE-2 and Edit distance, only the first text unit that is generated has a correct action associated with it. Later text units have generated tokens in their context, so there is no meaningful 'true' oracle action anymore. Hence, we only generate one text unit for the evaluation of ROUGE-2 and Edit distance in Figure 7.

We observe that all metrics do significantly better when the correct action is predicted, as expected. We can also see that increasing the fraction of planner-predicted actions during training improves incorrect perplexity, but damages correct perplexity. This can be explained by the fact that the model trained on fewer oracle actions learns to rely less on them. This effect is less consistent for the other metrics, although ROUGE-2 and Edit distance do tend to close the gap between examples with correct and incorrect actions when models train with a smaller fraction of oracle actions.

## F    Scalability and overheads of planner

**Overheads**    The planner introduces additional latency. Because the same planner pretraining stage can be reused for different language models, we focus our analysis on the latency introduced in the LM fine-tuning step. Specifically, we look at the time it takes to perform a forward and backward pass of one training batch, relative to the Fixed baseline. The Fixed baseline uses the same adapter parameters, but always selects the same single action embedding.

We compare three settings to the Fixed baseline:

- Uniform, which also doesn't use a planner, but computes the average of action embeddings

- Hard/Freeze, which uses the planner to select an action embedding, and does not train the planner during language model fine-tuning

- Soft/Unfreeze, which trains the planner during LM fine-tuning and uses a soft selection of action embeddings

Relative to Fixed, a training batch takes about 1.9 times longer for Uniform, about 3.9 times longer for Hard/Freeze, and about 7.2 times longer for Soft/Unfreeze. This is a significant overhead, but it is important to note that the planner is only invoked once per sentence, rather than for every token. Hence, it should be possible to significantly reduce this overhead by optimizing the implementation for speed. This is an important avenue for future work.

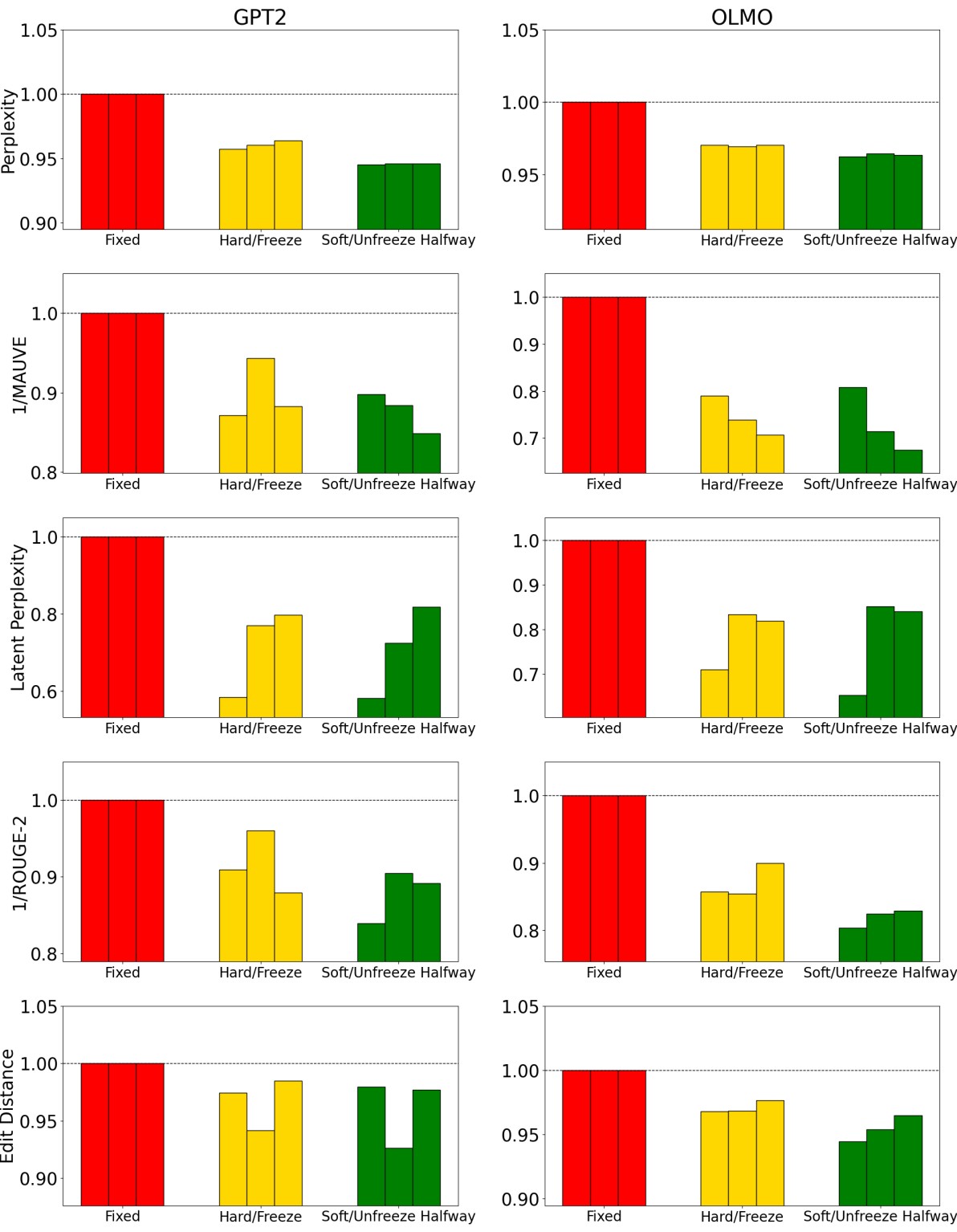

Figure 5: Relative metrics in different settings. Some metrics are inverted, so that lower is better for all metrics. Different bars of the same color indicate different random seeds. The random seed determines not only parameter initialization, but also which subsets of articles are used for training and evaluation. This means the inter-seed variance of absolute performance is significant. Because we are interested in the relative performance of different models, we show the metrics relative to the Fixed model with the same seed, which is scaled to 1 for each seed.

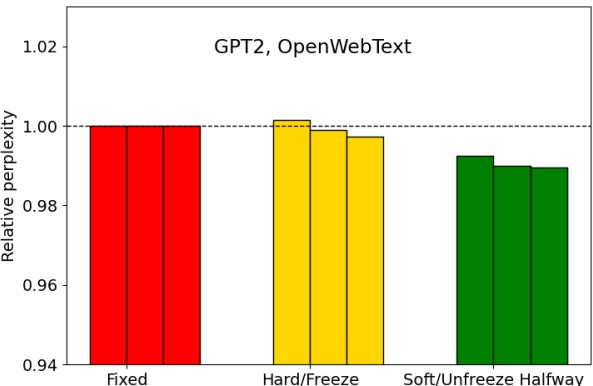

Figure 6: Relative perplexity in different settings for OpenWebText. As in Figure 5, different bars of the same color indicate different random seeds, and we show the perplexity relative to the Fixed model with the same seed, which is scaled to 1 for each seed.

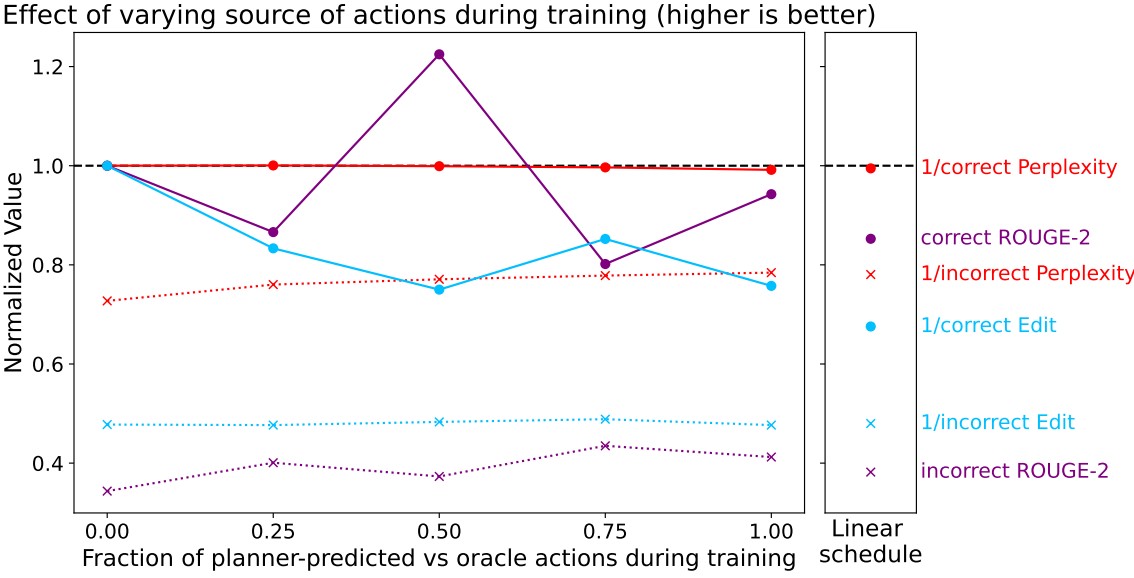

Figure 7: Relative improvement/worsening of our metrics as we increase the fraction of planner-predicted actions from zero (equivalent to Cornille et al. (2024) OA) to one (equivalent to Cornille et al. (2024) PA). We differentiate between examples in which the correct writing action was predicted (full lines), and examples in which it was not (dotted lines). Some metrics are inverted, so that higher is better for all metrics.

# End-to-end Planner Training for Language Modeling

**Anonymous authors**

## Abstract

Through end-to-end training to predict the next token, LLMs have become valuable tools for various tasks. Enhancing their core training in language modeling can improve numerous downstream applications. A successful approach to enhance language modeling uses a separate planning module to predict abstract labels of future sentences and conditions the LM on these predictions. However, this method is non-differentiable, preventing joint end-to-end tuning of the planner with the LM. We propose an effective method to improve this approach by enabling joint fine-tuning of the planner and the LM. We show that a naive way of approximating the gradient of selecting a label via the straight-through estimator is not effective. Instead, we propose to use the predicted label probabilities as mixing weights to condition the LM on a weighted average of label embeddings in a differentiable manner. This not only enables joint fine-tuning of the planner and the LM, but also allows the LM to draw on the full label distribution predicted by the planner, retaining more information. Our experimental results show consistent improvements in perplexity.

## 1 Introduction

Large Language Models (LLMs) currently lay the foundation for excellent performance across a variety of downstream tasks. They are not trained specifically for these tasks, but are pretrained in next token prediction, on trillions of tokens (typically followed by a short Reinforcement Learning from Human Feedback (RLHF) phase). Improving the effectiveness of the language modeling phase can be expected to lead to greater usefulness on many downstream tasks.

Cornille et al. (2024) are able to improve perplexity and generation quality by 1) pretraining a separate planning module to predict an abstract, discrete label of the next sentence (which they call "writing action", produced through unsupervised clustering) and 2) teaching the LM to condition on the planner's predictions while doing low-level next token prediction. This enables factorizing the language modeling task into prediction of abstract, high-level information processing and concrete, low-level generation conditioned on the high-level plan. Motivated by the idea of keeping the trained planner compatible with any LM, Cornille et al. (2024) do not fine-tune the planner during step (2). However, this means that the whole system is not trained end-to-end, breaking with one of the core concepts that make deep learning, and thus LLMs, so successful.

Therefore, the objective of this paper is to further improve LM performance by enabling end-to-end joint fine-tuning of the planner and the LM. This comes with two challenges: **(i)** The discrete choice of an action is non-differentiable. **(ii)** Useful high-level features that the planner has learned during step (1) may be forgotten due to the low-level objective in step (2).

In order to address challenge **(i)**, we first investigate the straight-through estimator as a method for approximating the gradient of the hard-selection of a writing action. Finding that this is not effective, we propose a method that uses the planner-predicted action probabilities to compute a weighted average of the action embeddings, rather than selecting an embedding based on which action has the highest probability. This has two advantages: First, it allows us to compute an exact gradient. Second, it does not discard valuable information contained in the label distribution predicted by the planner. We perform probing experiments that corroborate the hypothesis that using the full range of planner-predicted probabilities results in representations that are more informative about distant tokens.

To address challenge **(ii)**, we show two approaches that succeed at preventing catastrophic forgetting: one approach is to keep the planner parameters frozen for the first half of training before unfreezing them for the second half of training, another approach is to fine-tune the planner with a mix of its original Next-Action Prediction objective and the Next-Token Prediction objective. Moreover, we show that pretraining the planner with only the low-level objective (Next Token Prediction) damages performance, again underscoring the importance of the high-level objective (Next Action Prediction).

As observed in prior work, generation metrics can be improved by training the language model with oracle actions, as this means the language model learns to rely on the plan more strongly. However, this teacher forcing damages perplexity due to the mismatch of oracle actions during training with imperfect planner-predicted actions during evaluation, known as exposure bias. We show that we can strike a balance in this trade-off by mixing oracle and planner-predicted actions with a certain probability during training.

Our contributions can be summarized as follows:

- We propose a method that addresses the challenges of joint end-to-end training of a high-level planner module and a low-level language model.

- We compare our method using two LM backbones (GPT-2-small and OLMo-1B), demonstrating improvements in perplexity.

- We show that improvements depend on two key factors: (i) avoiding catastrophic forgetting of the planner's high-level knowledge—achieved by delaying the unfreezing of planner parameters or including the planner's high-level objective during training; and (ii) ensuring the LM has access to all planner-predicted probabilities rather than just the top prediction.

- We identify a trade-off between perplexity and generation quality that depends on the use of oracle or planner-predicted actions during training, and demonstrate that mixing oracle and planner-predicted actions balances this trade-off effectively.

## 2   Related Work

Our method is related to four fields of research. 1) Conditioning the language model on a writing action is akin to *Controllable Text Generation*, where a language model is conditioned on attributes of various types, such as content or style, to guide generation. 2) Since the planner operates at a higher level than the LM, our method is related to *Hierarchical Language Modeling.* 3) As the core contribution of our paper is enabling end-to-end training, we discuss techniques for *bridging the differentiability gap.*

**Conditioning language models on plans**   In Controllable Text Generation (Zhang et al., 2024), human-provided inputs are used to condition text generation, but we use latent, planner-produced inputs rather than human-provided ones. Chain-of-thought reasoning (Wei et al., 2023) improves outputs through explicit, token-level intermediate steps, and is not learned explicitly but only as a byproduct of LM pretraining. In contrast, the approach we build on conditions outputs on compact, latent-space plans, where a single plan embedding covers a sentence rather than individual tokens, and explicitly trains the model to condition generation on these plans. Quiet-STaR (Zelikman et al., 2024) conditions next token prediction on a model-produced input, but this is applied at every token, making it more computationally intensive, and is in text space rather than latent space. Wang et al. (2023) use planning tokens with a similar objective as the writing actions we consider, but they use an *internal* planner, i.e., the LM itself, to predict planning tokens, in contrast to the external planner from Cornille et al. (2024), who found an internal planner to be less flexible and less effective.

**Hierarchical Language Modeling**   Chung et al. (2017), Li et al. (2022), and Subramanian et al. (2020) all factorize language modeling into a per-token part and a slower 'per text-unit' part (e.g., per-phrase, per-sentence, or every fixed number of tokens). The per text-unit component in their work is only optimized for predicting concrete tokens. The key difference with this work is that our per text-unit component (i.e., the planner) is pretrained to predict targets in an abstract sentence space, and only then fine-tuned together

with the per-token component (i.e., the LM) to predict concrete tokens. Marfurt & Henderson (2021) and Jhamtani & Berg-Kirkpatrick (2020) also aim to improve language modeling by decomposing at the sentence and word level, but unlike Cornille et al. (2024) and this work do not operate at an *abstract* sentence level.

**Bridging differentiability gap**  A key component of our contribution is to enable the planner to be optimized jointly with the LM. A number of methods exist aimed at estimating a gradient for non-differentiable operations such as an argmax. Policy Gradient methods are one type of approach which are common in reinforcement learning (Williams, 1992; Sutton et al., 1999; Greensmith et al., 2004; Schulman et al., 2017). While unbiased, these tend to have high variance. Relaxed gradient estimators (Maddison et al., 2016; Paulus et al., 2020) constitute another approach, they replace a discrete sample with a continuous variable to calculate the gradient. An important variant of these is the straight-through estimator (Bengio et al., 2013), which uses a hard max for the forward pass, but a softmax for the backward pass. While this is a popular biased, but low-variance estimator of the gradient, we demonstrate that it yields unsatisfactory results in our scenario.

**Exposure bias and teacher forcing**  The method proposed by Cornille et al. (2024) learns a *cascaded model*, where the predicted class of the first model A (the planner) is propagated into the second model B (the language model). An old technique in such cases is to replace the actual prediction of model A with the ground truth signal (Jordan, 1986; Pineda, 1988). In sequence learning networks such as RNN this is known as *teacher forcing*, which can help stabilize training (Williams & Zipser, 1989). However, on the flip side, this approach can introduce an *exposure bias*, in which the distribution of inputs seen by model B at training time differs from inference time, leading to sub-optimal results (Bengio et al., 2015; Lamb et al., 2016). There are several ways to circumvent the exposure bias problem. The first type of approach aims to mitigate the training and test distribution mismatch. Scheduled sampling (Bengio et al., 2015) selects the self-predicted input with probability $p$ and the ground truth input with probability $1 - p$ during training. $p$ is scheduled to increase from 0 to 1 over the course of training to retain the best from both worlds. Professor Teaching (Lamb et al., 2016) uses an adversarial method to make the train and test distribution similar. Another type of approach circumvents the problem altogether by training the model end-to-end on the metric of interest, e.g. Ranzato et al. (2015) use self-predicted outputs in combination with reinforcement learning for text generation, and Graves & Jaitly (2014) directly optimizes for low word error rate in speech recognition. In this paper, we explore both scheduled sampling and end-to-end training to deal with the exposure bias.

## 3 Methodology

We summarize the common elements we preserve in 3.1. We then detail the prior way of interfacing the planner and the language model in 3.2, and our novel way in 3.3. We illustrate common elements and the contrast between the prior and novel component in Figure **??**.

### 3.1 Overview of Common Methodology

We consider the task of language modeling, where the goal is to estimate the probability $p(x_1 x_2 \ldots x_n)$ for any text sequence $X = x_1 \ldots x_n \in \mathcal{X}$. We also refer to this task as *Next Token Prediction*. The probability is factorized into the product of probabilities of each token given the preceding tokens: $p(x_1 \ldots x_n) = \prod_{i=1}^{n} p(x_i | x_1 \ldots x_{i-1})$. Unlike a standard LM, the predicted probability in our method is conditioned on additional predictions by an external planner.

To obtain training data for this planner, we apply a pretrained text encoder (e.g., Sentence-BERT) to each sentence $t_j$ in the corpus ~~is first embedded into a~~ to produce a corresponding low-dimensional vector $\mathbf{z}_j$ ~~using some pretrained text encoder (e.g., Sentence-BERT)~~. K-means clustering on all embedded text units from the corpus yields clusters that are used as abstract labels (or 'writing actions') $a \in \mathcal{A}$. A cluster's centroid serves as the (initial) action embedding. This is shown in step 1 in Figure **??**.

The planner module is then pretrained to predict the writing action $a_i$ that corresponds to sentence $t_i$ based on the context of the preceding text units $t_1, \ldots, t_{i-1}$. This task is called *Next Action Prediction*. The

planner module is based on a custom Transformer architecture that first embeds each sentence independently into a single vector and then contextualizes them with a Transformer encoder. This is shown in step 2 in Figure **??**. For more details, please refer to Cornille et al. (2024).

### 3.2 Prior approach to Language Model Fine-tuning

In Cornille et al. (2024), the input $X$ is split into $m$ sentences $X = t_1 \ldots t_m$, where each sentence $t_j = x_1^j \ldots x_{n_j}^j$ has a single associated *predicted* writing action $\hat{a}_j$, predicted by a pretrained planner. The fine-tuning objective then is to estimate $\prod_{j=1}^m \prod_{i=1}^{n_j} p(x_i^j | \hat{a}_1 \ldots \hat{a}_j, t_1 \ldots t_{j-1} x_1^j \ldots x_{i-1}^j)$.

The predicted actions are one-hot encoded, and used as index to select action embeddings in the action embedding matrices $E_A^l$ that are added at various layers $l$ of the LM, resulting in a vector $r_j^l = E_A^l(\hat{a}_j)$. That vector is then projected and element-wise added to each hidden representation in layer $l$, just after the multiplication with attention weights and just before the output projection, similar to Llama-Adapter (Zhang et al., 2023).

Importantly, planner parameters are not updated during LM fine-tuning, disallowing the planner to improve its action predictions beyond matching cluster labels, by tailoring them to the LM.

### 3.3 Novel Joint Planner-Language Model Fine-tuning

We hypothesize that the planner can enhance the utility of its predictions by being fine-tuned for next-token prediction, jointly with the LM, after being pretrained for Next Action Prediction. Hence, we want to enable the gradient to be passed into the planner.

A naive way to achieve this is using a straight-through estimator, which involves using a hardmax function for the forward pass but a softmax function for the backward pass

$$\text{onehot}(\text{argmax}(\mathbf{s}))$$

during gradient computation:

$$\text{hardmax}(\mathbf{s}) = \text{onehot}(\text{argmax}(\mathbf{s})) \tag{1}$$

$$\left[ \frac{e^{s_1}}{\sum_{j=1}^{|A|} e^{s_j}}, \frac{e^{s_2}}{\sum_{j=1}^{|A|} e^{s_j}}, \ldots, \frac{e^{s_{|A|}}}{\sum_{j=1}^{|A|} e^{s_j}} \right]$$

$$\text{softmax}(\mathbf{s}) = \left[ \frac{e^{s_1}}{\sum_{j=1}^{|\mathcal{A}|} e^{s_j}}, \frac{e^{s_2}}{\sum_{j=1}^{|\mathcal{A}|} e^{s_j}}, \ldots, \frac{e^{s_{|\mathcal{A}|}}}{\sum_{j=1}^{|\mathcal{A}|} e^{s_j}} \right] \tag{2}$$

Here, $\mathbf{s} = [s_1, \ldots, s_{|A|}]$

Here, $\mathbf{s} = [s_1, \ldots, s_{|\mathcal{A}|}]$ is the vector of planner-predicted logits for each of $|\mathcal{A}|$ possible actions, and onehot($i$) returns a vector of length $|\mathcal{A}|$ where the $i$-th element is 1 and the rest are 0 (one-hot encoding).

In the forward pass, we compute the action selection using the hardmax function (Eq. 1), which produces a one-hot encoded vector indicating the action with the highest logit. However, during backpropagation, we use the softmax function (Eq. 2) to approximate the gradients, effectively allowing gradients to flow through the non-differentiable argmax operation.

The straight-through estimator is limited by being a biased estimator of the gradient however. A more effective method arises from the insight that there is no inherent need to select only one action. Instead, we can use the planner probabilities to obtain a weighted average of the action embedding columns:

$\mathbf{r}_j^l = \sum_{a \in \mathcal{A}} *\theta(\mathbf{s})_a \cdot E_A^l(a), \mathbf{r}_j^l = \sum_{a \in \mathcal{A}} \text{softmax}(\mathbf{s})_a \cdot E_A^l(a),$ Step 3 in Figure ?? illustrates this approach. This approach not

The purpose of our experiments is twofold. First, we want to test our hypothesis that end-to-end joint planner-LM training is beneficial for language modeling performance. Second, we want to validate the design decisions we made to enable end-to-end training: a) Using a soft-selection via weighted average rather than a hard selection, and b) mitigating catastrophic forgetting by unfreezing the planner only after half the training or using a mixed Next Action / Next Token prediction objective.

### 4.1 Dataset and backbone models

We train and evaluate our models on the same dataset as Cornille et—, i.e., subsets of English Wikipedia articles from the "20220301.en" version from Huggingface

**Primary evaluation** Our main metric is perplexity, which is the default metric used for language modeling, corresponding to the inverse geometric mean of the probability of true texts according to the language model.

**Generation evaluation** As in Cornille et al. (2024), we complement perplexity, which does not directly assess generated text, with generation metrics. We report ROUGE-2 (F1) (Lin, 2004) and MAUVE (Pillutla et al., 2021) to evaluate generated texts at the surface level, and Levenshtein distance (Levenshtein et al., 1966) and latent perplexity (Deng et al., 2022) to assess text quality at an abstract level. For the surface level, ROUGE-2 evaluates bigram overlap between generated and real text, while MAUVE measures the divergence between model and true data distributions by comparing generated and real texts unconditionally. For the abstract level, we first map true and generated texts onto the sequence of writing actions that correspond to them. Levenshtein distance then measures the edit distance between generated and ground-truth writing action sequences, and latent perplexity estimates how well the generated sequence aligns with a latent HMM-based critic trained on real texts. We refer to appendix B for more details about the generation evaluation.

**Probing** In order to understand how the different training setups influence what information the model (un)learns, we use probing classifiers on top of the (frozen) representations to determine how well they predict the upcoming tokens. The choice of probing classifier is not straight-forward (Belinkov, 2022). We choose linear probing classifiers to measure to what extent the information about upcoming tokens can be easily extracted (i.e., is linearly separable) from the representations, rather than be learned by the probe itself (Alain, 2016).

We probe representations at two kinds of locations inside the model. First, we probe the output from the action embedding inside the adapter, which contains information only from the planner (Pre-merge). Second, we probe the representation after the information of the planner has been mixed with the contextualized information from the LM itself (Post-merge). We train probes at every layer where the planner information is infused.

Figure 2 illustrates ~~this~~ the two kinds of probing locations.

### 4.3 Settings

**Variations of end-to-end planner** We evaluate the impact of 4 properties of the end-to-end planner.

First, whether the planner's parameters are unfrozen immediately (*Unfrz immediate*), halfway through training (*Unfrz halfway*), or never (*Unfrz never*). We expect that immediately unfreezing the

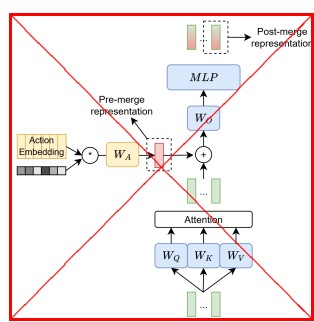

Figure 2: Illustration of the probing locations inside the model at one layer. The "Action embedding" and $W_A$ are the parameters of the Action Adapter at that layer, $W_A$ is a linear projection matrix. $W_Q$, $W_K$, $W_V$ and $W_O$ are the query, key and value projection

planner when the LM hasn't adapted to it yet might lead to catastrophic forgetting, while not unfreezing it at all doesn't allow the planner to tune itself to the LM.

Second, similarly aimed at preventing catastrophic forgetting, we evaluate the effect of continuing to train the planner for its Next-Action Prediction objective at the same time as also tuning end-to-end for Next-Token Prediction Objective.

Third, we evaluate the effect of replacing soft-selection via weighted averages (Eq. 3.3) with hard-selection via the straight-through estimator (Eq. 1 and 3.3).

Finally, because we are now able to train the planner end-to-end, we evaluate whether its Next-Action-Prediction (NAP) pretraining objective is still necessary by assessing models in which the planner is only pretrained with a Next-Token-Prediction objective. Specifically, we replace NAP training of the planner with an end-to-end stage in which we keep the LM parameters frozen.

**Baselines**   As baseline without any planner, we follow Cornille et al. (2024) in evaluating the model that includes the same additional adapter parameters that are fine-tuned, but always selects the same fixed action embedding, rather than relying on a planner to select action embeddings (*Fixed*).

To rule out that the benefit of ~~soft-mixing~~ soft-selection is not merely due to mixing multiple actions, we train a variant of the soft-selection method that always applies uniform weighting across all actions (*Uniform*).

Our main baselines are the planner models proposed in Cornille et al. (2024). They have two variants: one pretrained on oracle actions (OA), and one pretrained on predicted actions (PA). Cornille et al. (2024) observed a trade-off between these variants: while PA had better perplexity, OA performed better in some generation metrics. We explore this trade-off more in-depth by making models that mix oracle-action and predicted actions during ~~pretraining~~ fine-tuning of the language model.

## 5   Results and Discussion

### 5.1   Main results

Table 1 shows our main results.

**Benefit of end-to-end training**   Comparing **Ours (soft-selection)** to the **Baselines**, the results confirm our hypothesis that end-to-end joint planner-LM training can improve language modeling performance compared to the prior approach, with our best setting improving by 0.3 (GPT-2) and 0.08 perplexity (OLMo), respectively over *Cornille et al. (2024) PA* (Predicted Actions).

We observe that this perplexity improvement does not always translate into improved generation metrics. As noted in Cornille et al. (2024), there is a trade-off between perplexity and performance on generation metrics stemming from the use of teacher forcing for the actions. We examine this trade-off in more detail in section 5.3.

Using soft-selection and end-to-end training does introduce some additional latency, whch we discuss in Appendix F.

**Soft selection beats hard selection**   Comparing **Ours (soft-selection)** to **Ours (straight-through)**, we see that soft-selection variants are consistently better than straight-through variants. This can be explained by two factors. First, the biased gradient estimates of the straight-through estimator might lead the learning astray. Second, soft-selection has the benefit of allowing the LM to draw on the full label distribution: In fact, the soft-selection *Unfrz never* result shows that this alone is already beneficial, even without updating the planner. This explanation is corroborated by the probing results presented in section 5.2, which

| Base LM | | GPT2 | | | | | OLMO | | | |
|---|---|---|---|---|---|---|---|---|---|---|
| Setting | PPL ↓ | MAUVE ↑ | Latent PPL ↓ | ROUGE-2 ↑ | Edit ↓ | PPL ↓ | MAUVE ↑ | Latent PPL ↓ | ROUGE-2 ↑ | Edit ↓ |
| **Baselines** | | | | | | | | | | |
| Cornille et al. (2024) OA | 26.94 | 0.447 | 91.60 | 0.0193 | 3.69 | 11.99 | 0.411 | 76.20 | 0.0278 | 3.26 |
| Cornille et al. (2024) PA | 25.55 | 0.435 | 205.90 | 0.0169 | 3.78 | 11.46 | 0.563 | 178.20 | 0.0253 | 3.31 |
| Fixed | 26.69 | 0.379 | 352.80 | 0.0154 | 3.88 | 11.81 | 0.445 | 250.90 | 0.0217 | 3.42 |
| Uniform | 26.69 | 0.378 | 354.27 | 0.0159 | 3.91 | 11.81 | 0.396 | 256.13 | 0.0219 | 3.43 |
| **Ours (soft-selection)** | | | | | | | | | | |
| Unfrz immediate | 25.42 | 0.423 | 245.48 | 0.0178 | 3.68 | 11.42 | 0.564 | 187.61 | 0.0271 | 3.33 |
| Unfrz halfway | 25.23 | 0.422 | 205.14 | 0.0183 | 3.80 | 11.37 | 0.551 | 163.78 | 0.0270 | 3.23 |
| Unfrz never | 25.32 | 0.420 | 187.54 | 0.0175 | 3.74 | 11.49 | 0.546 | 163.81 | 0.0271 | 3.29 |
| **Ours (straight-through)** | | | | | | | | | | |
| Unfrz immediate | 25.94 | 0.401 | 281.34 | 0.0162 | 3.87 | 11.53 | 0.548 | 208.12 | 0.0229 | 3.36 |
| Unfrz halfway | 25.66 | 0.464 | 230.00 | 0.0171 | 3.79 | 11.42 | 0.547 | 185.76 | 0.0254 | 3.34 |
| **Ours (NAP during fine-tuning)** | | | | | | | | | | |
| Unfrz immediate | 25.24 | 0.459 | 177.34 | 0.0172 | 3.72 | 11.42 | 0.576 | 155.38 | 0.0266 | 3.29 |
| **Ours (no NAP pretraining)** | | | | | | | | | | |
| Unfrz immediate | 25.80 | 0.443 | 271.71 | 0.0167 | 3.83 | 11.69 | 0.562 | 227.46 | 0.0231 | 3.44 |
| Unfrz halfway | 25.82 | 0.397 | 299.62 | 0.0165 | 3.78 | 11.66 | 0.534 | 224.99 | 0.0218 | 3.37 |
| Unfrz never | 26.15 | 0.435 | 299.51 | 0.0163 | 3.78 | 11.80 | 0.403 | 258.24 | 0.0204 | 3.44 |

Table 1: Perplexity and generation metrics under different training and conditioning scenarios. Cells shaded in red show perplexity, those in blue show the generation metrics. A darker color indicates a worse result.

show that linear probes trained on the soft-selected planner output (rather than the hard-selected one) are better able to predict distant tokens.

Soft-selection also activates the full action embedding matrix at every prediction. However, the fact that *Uniform* performs considerably worse shows that just using the full embedding matrix is *not* responsible for the improvement.

**Timing matters for planner unfreezing**  Keeping the planner frozen during part of the training is more effective than either immediately unfreezing the planner or keeping it frozen the entire time. This is in line with our hypothesis that immediately unfreezing the planner leads to big initial gradients that erase some of the useful knowledge built up during the Next-Action-Prediction planner pretraining phase. On the other hand, not unfreezing the planner at all prevents the planner parameters from specializing toward perplexity minimization.

Our alternative approach to preventing catastrophic forgetting (**Ours (NAP during fine-tuning)**) achieves performance nearly on par with unfreezing the planner halfway.

**Next-Action-Prediction objective cannot be left out completely**  Finally, we see that the models we run with **no NAP pretrainnig** are generally worse for both perplexity and generation metrics than **Ours (soft-selection)**. This indicates that the abstract pretraining objective of the planner is still required, even when end-to-end training is possible.

## 5.2 What do the models learn?

Figure 3a shows the results of our probing experiments by distance to the target token. Unsurprisingly, tokens farther away tend to be more difficult to predict. Regarding pre-merge representations, the Cornille et al. (2024) baseline is notably worse than our proposed methods using ~~softmix~~ soft-selected representations, which benefit from making better use of the full planner's predicted scores rather than only the argmax. Generally, the post-merge representations are significantly better than the pre-merge representations. In fact, the language model alone, without any (useful) planner information ("Uniform") already predicts farther into the future than just the next token. However, adding the pre-merge representations of the planner yields further improvements. Moreover, freezing the pretrained planner at least for half the training epoch tends to retain more information about the upcoming tokens than unfreezing it immediately.

While this probing experiment cannot prove a causal mechanism, it is plausible that the improved performance observed in Table 1 is at least partially attributable to the models ability to being better at predicting several tokens ahead.

Figure 3b shows that the information contained in pre-merge representations is largely independent of the layer, which is explained by the fact that lower layer representations do not feed into higher layer representations. In contrast, post-merge results clearly show that higher layers, which are located closer to the output layer that performs the final token prediction, contain more information about the upcoming tokens.

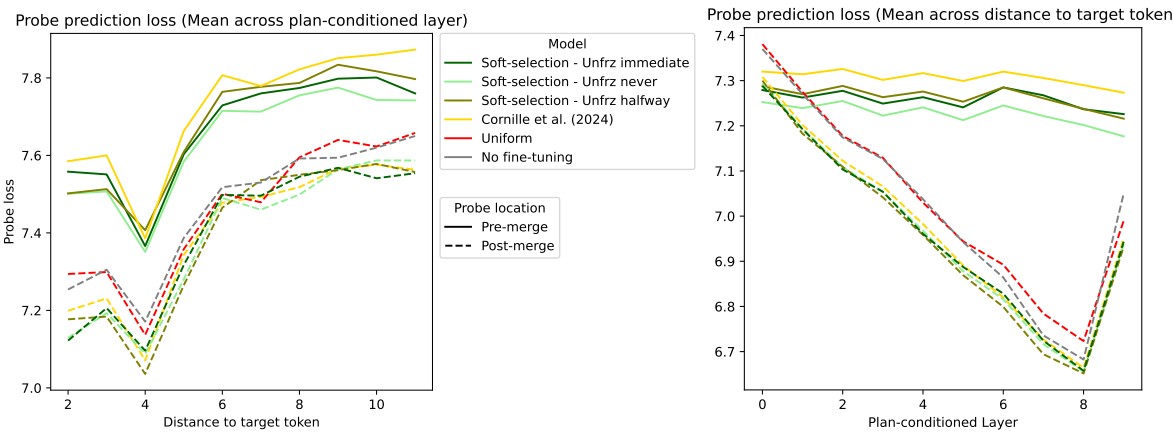

(a) Probe Layer Mean, for tokens at least 2 positions into the future.    (b) Probe Target Mean

Figure 3: Plots showing probing performance at different layers and for different distances to the probe's target token. The distance to target token indicates how many positions into the future the target token is relative to the token immediately following the Language Model input.

## 5.3    Trade-off between Perplexity and Generation Metrics

To investigate the trade-off between perplexity and generation metrics (MAUVE, ROUGE-2, Edit distance and Latent Perplexity), we train models that use a mixture of oracle and planner-predicted actions during training, where we vary the fraction of planner-predicted actions from 0 (equivalent to *Cornille et al. (2024) OA*) to 1 (equivalent to *Cornille et al. (2024) PA*). The left side of Figure 4 shows that the smaller the fraction of oracle actions during training, the better the perplexity, up to an improvement of around 5%. Because perplexity evaluation happens with planner-predicted actions, the bigger the fraction of oracle actions during training, the bigger the training/evaluation mismatch, a problem known as exposure bias.

The perplexity improvement does not translate into improving generation metrics, with some metrics (Latent Perplexity and ROUGE-2) even consistently worsening. To understand this, consider the plan-matching accuracy (green line). As fewer actions are oracle, the plan-matching accuracy decreases, indicating the language model learns to rely less on the plan. This suggests that these generation metrics benefit from having a model rely more on the planner output, even if it is imperfect.

To try to get the best of both worlds, we also evaluate a setting with a scheduled fraction that linearly increases the fraction of planner-predicted actions from 0 to 1 during training, i.e., scheduled sampling, shown on the right side of Figure 4. However, we observe that this leads similar results as training only on planner-predicted actions.

Hence, overcoming this trade-off by both overcoming the problem of exposure bias and ensuring the language model learns to sufficiently rely on the proposed plans is an interesting avenue for future work.

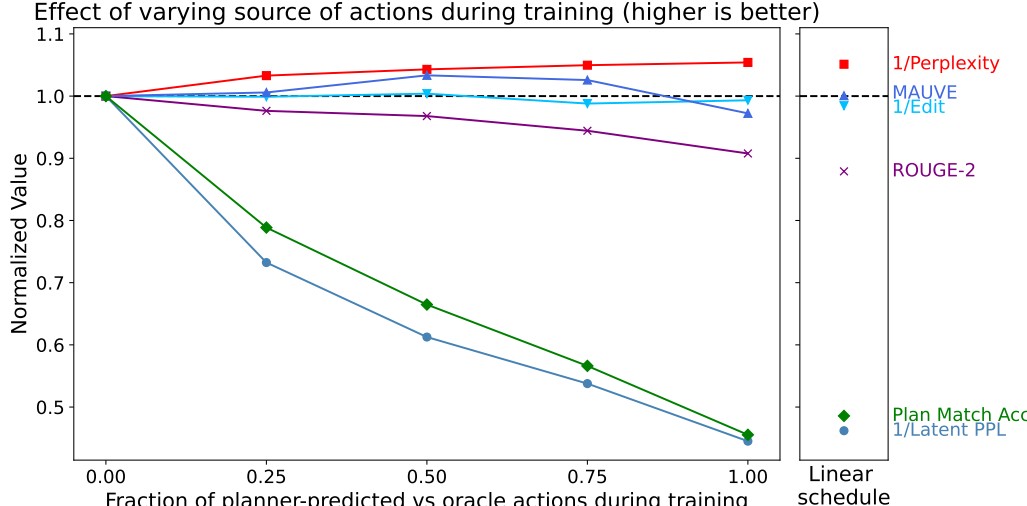

Figure 4: Relative improvement/worsening of our metrics as we increase the fraction of planner-predicted actions from zero (equivalent to Cornille et al. (2024) OA) to one (equivalent to Cornille et al. (2024) PA). Some metrics are inverted, so that higher is better for all metrics.

In Appendix E, we show the same model settings as in Figure 4, but show separate metrics for test examples in which the correct writing action was predicted, and examples in which it was not. As expected, we find that test examples with correct actions score much better.

## 6 Conclusion

Since end-to-end training is a key ingredient to the success of deep learning, it is important that we enable different system components to be optimized jointly. In this work, we bridge the differentiability gap of a recent pretrained planning module with a language model by turning the indifferentiable hard-selection into a differentiable soft-selection. Our results demonstrate that this consistently improves perplexity. We hope these findings can provide a foundation for enhancing production-scale language models through end-to-end planning mechanisms.

# 7 Limitations

## 7.1 Model Size

Due to computational constraints, our evaluation was performed on relatively small models. Consequently, the scalability and effectiveness of the proposed method need to be validated on production-scale models to ensure its applicability in real-world scenarios.

## 7.2 Planning Horizon

Our approach involves planning only one step into the future. This is a simplification compared to how humans presumably think and plan farther into the future. Future work should investigate methods to extend the planning horizon, allowing the model to consider multiple future steps and thereby improve decision-making processes.

### Broader Impact Statement

While increasingly more capable LLMs are very useful, they can also be misused for harmful purposes (such as generating disinformation, helping in development of weapons, etc.). Because our work has used LLMs of modest size, there is little risk of it contributing to such misuses directly. It could do so however if our method would be used to make production-scale language models even more effective. If that is the case, it is important to take the necessary precautions before deployment, such as proper alignment with human values.

The compute requirements of large models also have a significant environmental impact (Rillig et al., 2023). Use of a planning module also entails additional compute requirements, which can further contribute to this, although the planning module is relatively lightweight compared the the Language Models, and is invoked only once per sentence rather than for every token.

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

## A    Model Details

**Parameter counts**   Table 2 shows parameter counts for our models

| Model | Parameter Count |
|---|---|
| GPT2-Small | 124,439,808 |
| Olmo | 1,176,764,416 |
| Extra conditioning parameters | 13,770,240 |
| Planner parameters | 116,378,496 |

Table 2: Parameter counts for our models

**Computational Budget**   We ran our experiments on either 12GB, 16GB or 24GB GPUs, each time using one GPU per experiment. We report a single run for each setting. With this setting, joint fine-tuning of planner and LM takes around 40 hours for GPT2-Small and 60 hours for Olmo 1B. Pretraining the planner for Next Action Prediction took around 90 hours, but we reuse the same pretrained planner for most experiments. Evaluating perplexity takes about 3-5 hours, while evaluating edit-distance (which requires generation) takes around 10-15 hours.

We estimate that we ran about 100 experiments (only a subset of which led to results presented in the paper), which means in total we used around 7000 GPU hours.

**Used artifacts**   We build on the source code of Cornille et al. (2024), which was shared with us privately. We will release our extensions publicly once their code is made available.

We use spaCy (Honnibal et al., 2020) to split articles into sentences. These sentences are then transformed into embeddings using MPNet-base-v2 (Song et al., 2020) through the SentenceTransformer library (Reimers

& Gurevych, 2019)[3]. The final clustering is conducted using k-means++ initialization (Arthur & Vassilvitskii, 2007) implemented in Scikit-Learn (Pedregosa et al., 2011).

The Wikipedia dataset is accessed via the 'datasets' library at `https://huggingface.co/datasets/wikipedia`, specifically the March 2022 version ('20220301').

We use PyTorch (Paszke et al., 2019), the Huggingface 'datasets' (Lhoest et al., 2021), and 'transformers' (Wolf et al., 2020) libraries for loading and preprocessing data and pretrained models (specifically GPT-2 (Radford et al., 2019)). Additionally, we employ PyTorch-Lightning (Falcon et al., 2020) for model training. All the libraries utilized are open source or freely available for academic use.

**Hyperparameters**   Table 3 shows hyperparameters used for our experiments. We do not perform hyperparameter search for these, using the default hyperparameters reported in Cornille et al. (2024). We use the Adam optimizer (Kingma, 2014), and always train for one epoch.

Table 3: Hyperparameter Settings

| Hyperparameter | Value |
|---|---|
| Context window size | 128 |
| Train \| test \| val split sizes | 285310 \| 1000 \| 1000 |
| K-means initialization | k-means++ |
| Default action count | 1024 |
| Action embedding dimension | 768 |
| **Language Model Fine-tuning** | |
| Batch size | 32 |
| Learning rate | 1e-4 |
| **Planner Training** | |
| Batch size | 32 |
| Learning rate | 1e-4 |

## B   Generation Evaluation Setup and Detailed Results

We follow the evaluation setup from Cornille et al. (2024), and explain the details again here:

For MAUVE and Latent Perplexity, we generate 1024 tokens unconditionally (i.e., without context), matching the average length of the articles in the dataset.

For ROUGE-2, and Edit distance, we use a prefix $t_1 \ldots t_i$ from real texts and generate continuations from that prefix of 128, 256, 512, and 1024 tokens. Because Edit distance scales linearly with the number of tokens, we normalize the results across different lengths. For 128 tokens, we report the raw edit distance; for 256, we divide the edit distance by 2, and so on, ensuring a consistent comparison across generation lengths.

The results in the main text (Table 1) are the average for these different generation lengths.

## C   Repeat experiments with different random seeds

Figure 5 compares performance for multiple random seeds. Three models are compared:

- Fixed: a baseline that always receives the same, uninformative, plan

---

[3]`https://huggingface.co/sentence-transformers/all-mpnet-base-v2`

- Hard/Freeze: the model that achieved the best perplexity from Cornille et al. (2024), which uses a planner that selects a single writing action, and doesn't fine-tune the planner parameters towards next token prediction

- Soft/Unfreeze Halfway: the best model from this paper, which uses a mix of writing actions based on planner-predicted probabilities, and unfreezes the planner after some time to be fine-tuned for next token prediction.

It shows that in perplexity, Hard/Freeze consistently improves over Fixed, and that Soft/Unfreeze Halfway consistently improves over Hard/Freeze. For other metrics, models with a planner (Hard/Freeze and Soft/Unfreeze Halfway) are consistently better than Fixed. However, Soft/Unfreeze Halfway does not consistently improve generation metrics compared to Hard/Freeze. We hypothesize this is connected with the trade-off due to plan-matching discussed in section 5.3. Ensuring that the end-to-end planner perplexity benefits translate into improved generation metrics too is an import avenue for future work.

## D    Performance on OpenWebText

For our main experiments, we evaluate on Wikipedia articles because they are structured and likely to benefit from a high-level planner.

In this appendix, we show results for an additional dataset, namely OpenWebText (Gokaslan et al., 2019), an open-source replication of the WebText dataset from OpenAI, that was used to train GPT-2. It covers a broader range of data sources than only Wikipedia articles.

The results are shown in Figure 6.

We observe that the planner proposed by Cornille et al. (2024) does not consistently improve over the baseline for the OpenWebText dataset, while our improved planner does. The improvement over the Fixed baseline is significantly smaller however. We hypothesize that this is because this dataset is more varied and less structured than Wikipedia articles, requiring a more powerful planner with a larger action space trained on significantly more data. We leave this for future work.

## E    Effect of correct versus incorrect action predictions for varying mix of oracle and predicted codes

Figure 7 shows separate metrics for test examples in which the correct writing action was predicted, and examples in which it was not, for setting that differ in their mixture of oracle and planner-predicted actions during training.

Note that for MAUVE and Latent Perplexity, there is no oracle action, because these metrics don't compare individual generations to matching true texts, but groups of generated texts to groups of true texts. Further, for ROUGE-2 and Edit distance, only the first text unit that is generated has a correct action associated with it. Later text units have generated tokens in their context, so there is no meaningful 'true' oracle action anymore. Hence, we only generate one text unit for the evaluation of ROUGE-2 and Edit distance in Figure 7.

We observe that all metrics do significantly better when the correct action is predicted, as expected. We can also see that increasing the fraction of planner-predicted actions during training improves incorrect perplexity, but damages correct perplexity. This can be explained by the fact that the model trained on fewer oracle actions learns to rely less on them. This effect is less consistent for the other metrics, although ROUGE-2 and Edit distance do tend to close the gap between examples with correct and incorrect actions when models train with a smaller fraction of oracle actions.

## F    Scalability and overheads of planner

**Overheads** The planner introduces additional latency. Because the same planner pretraining stage can be reused for different language models, we focus our analysis on the latency introduced in the LM fine-tuning step. Specifically, we look at the time it takes to perform a forward and backward pass of one training batch, relative to the Fixed baseline. The Fixed baseline uses the same adapter parameters, but always selects the same single action embedding.

We compare three settings to the Fixed baseline:

- Uniform, which also doesn't use a planner, but computes the average of action embeddings
- Hard/Freeze, which uses the planner to select an action embedding, and does not train the planner during language model fine-tuning
- Soft/Unfreeze, which trains the planner during LM fine-tuning and uses a soft selection of action embeddings

Relative to Fixed, a training batch takes about 1.9 times longer for Uniform, about 3.9 times longer for Hard/Freeze, and about 7.2 times longer for Soft/Unfreeze. This is a significant overhead, but it is important to note that the planner is only invoked once per sentence, rather than for every token. Hence, it should be possible to significantly reduce this overhead by optimizing the implementation for speed. This is an important avenue for future work.

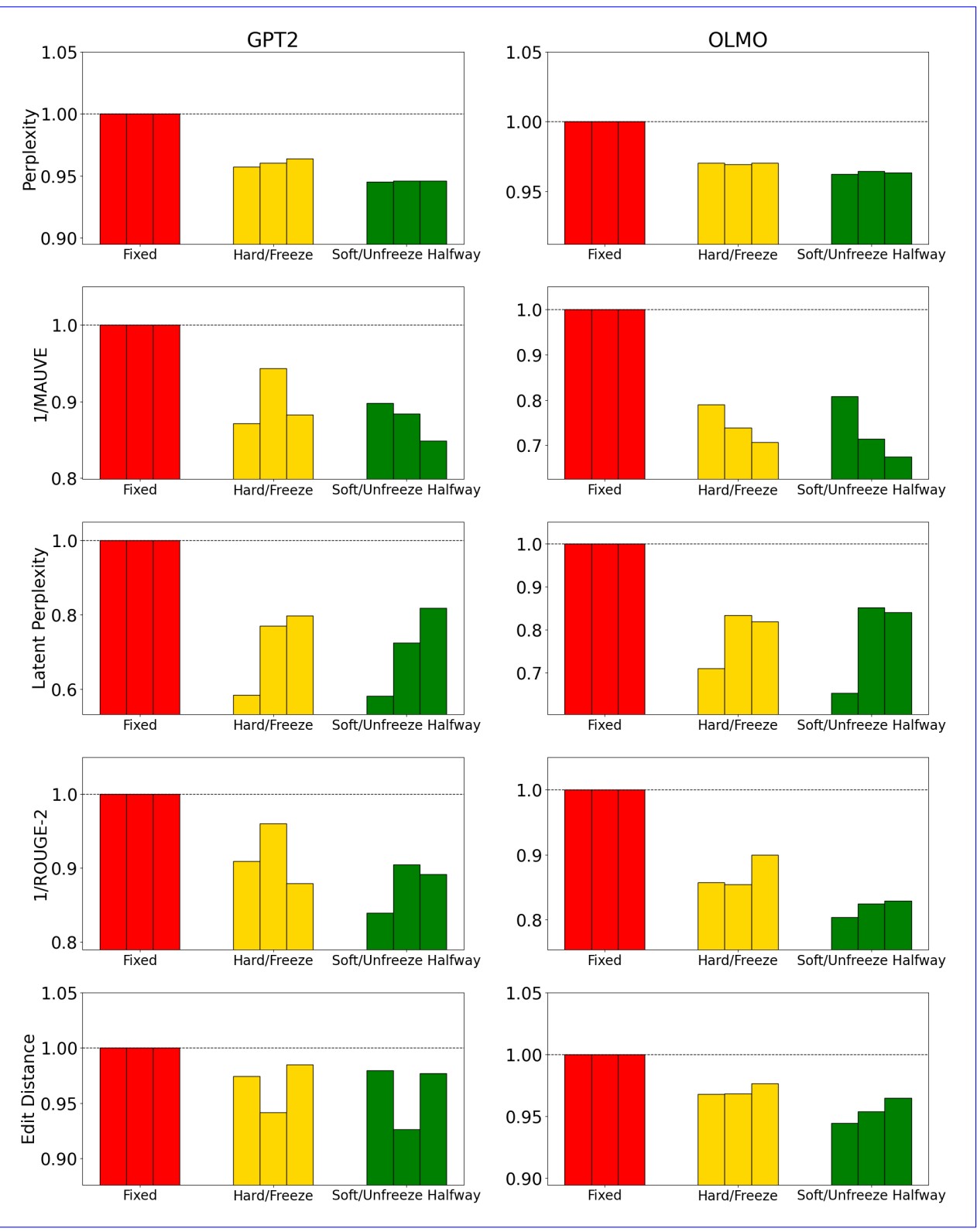

Figure 5: Relative metrics in different settings. Some metrics are inverted, so that lower is better for all metrics. Different bars of the same color indicate different random seeds. The random seed determines not only parameter initialization, but also which subsets of articles are used for training and evaluation. This means the inter-seed variance of absolute performance is significant. Because we are interested in the relative performance of different models, we show the metrics relative to the Fixed model with the same seed, which is scaled to 1 for each seed.

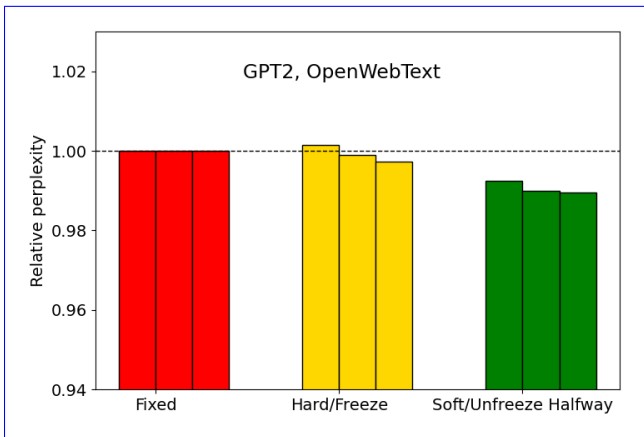

Figure 6: Relative perplexity in different settings for OpenWebText. As in Figure 5, different bars of the same color indicate different random seeds, and we show the perplexity relative to the Fixed model with the same seed, which is scaled to 1 for each seed.

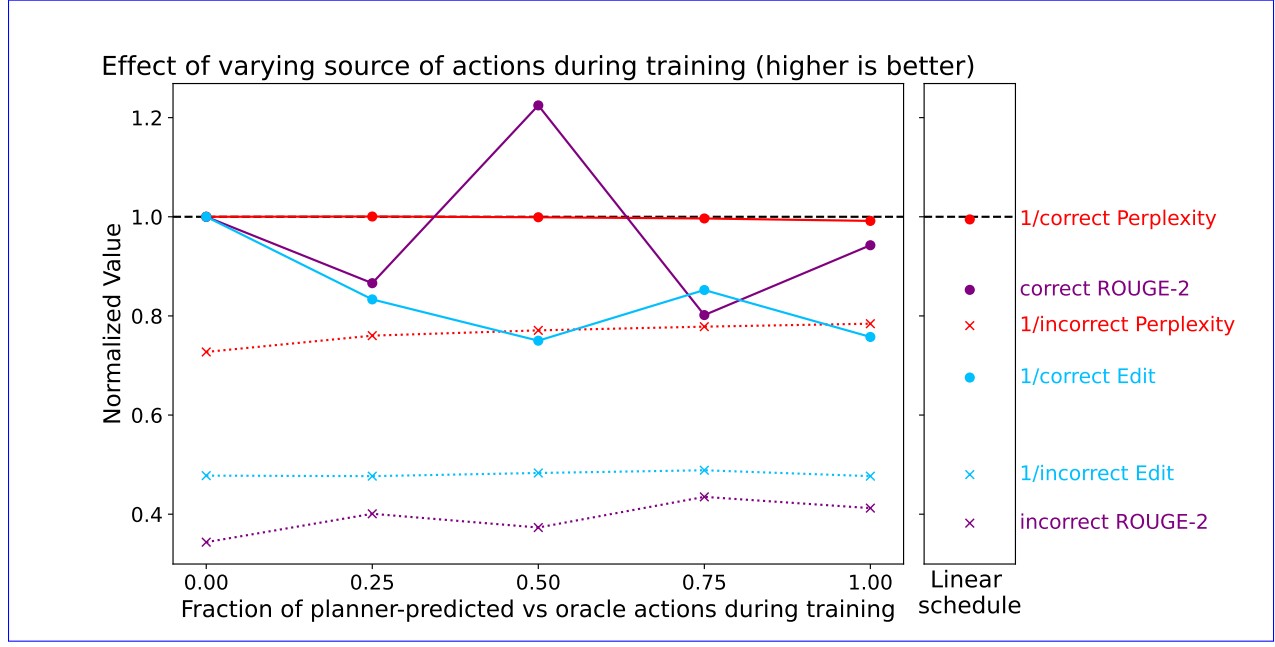

Figure 7: Relative improvement/worsening of our metrics as we increase the fraction of planner-predicted actions from zero (equivalent to Cornille et al. (2024)OA) to one (equivalent to Cornille et al. (2024)PA). We differentiate between examples in which the correct writing action was predicted (full lines), and examples in which it was not (dotted lines). Some metrics are inverted, so that higher is better for all metrics.

