# OpenReview forum: "End-to-end Planner Training for Language Modeling"
_TMLR — Rejected by TMLR_

### Review · Reviewer_3NNJ · 2024-11-03

**Summary Of Contributions:**

A new end-to-end training pipeline is presented for LLMs, wherein an additional planner module is jointly trained with the LLM and used to condition the LLM's predictions. Key contributions are: (1) The use of soft planner probabilities (and actions) during training to enable differentiability, and (2) The use of different tricks -- freezing planner parameters during initial fine-tuning iterations and training using both Next Action Prediction and Next Token Prediction -- to avoid catastrophic forgetting.

**Audience:**

Yes

**Broader Impact Concerns:**

Sufficiently addressed

**Claims And Evidence:**

No

**Requested Changes:**

Critical for acceptance:
+ Quantify the training and inference overheads of using the planner module, in terms of latency and memory requirements. It looks like the planner adds additional training and inference latency, since both the planner and the original LLM now need to be evaluated.
+ Describe how planner size and latency change with LLM architecture. Do planner size and latency increase as LLM size increases, or can the same planner module be used irrespective of the underlying LLM architecture?
+ Add details on how the optimal hyperparameters for training the planner can be found when training on a new task or LLM. In particular, how can users determine when to unfreeze planner parameters during training, and how often (and during which iterations) to use next action prediction vs next token prediction objective. Can this process be automated? How long does this process take?
+ Add a new row to Table 1 with results on vanilla training and fine-tuning without the use of planner. The improvements from the planner, along with its overheads= numbers, will help better assess the significance of the work


Would strengthen the paper:
+ The paper is not self-contained in its current form. Some important details, such as the planner model architecture, are only present in the references, making it difficult for readers unfamiliar with the field. It would be good to add all necessary details required to reproduce this work in the paper, either in the main section or in the appendix

**Strengths And Weaknesses:**

Strengths:
+ As LLMs are rapidly growing in popularity, this paper makes a timely contribution that can potentially improve generation quality
+ Writing is clear (although some important details are missing, see Weaknesses below), and the methods proposed in the paper are clearly described
+ Experiments presented show that the proposed techniques can indeed improve planner quality


Weaknesses:
+ The need for the planner module, and its advantages are unclear. In fact, the authors do not include any results in the paper showing that the incorporation of the planning module does indeed improve generation quality over vanilla training and fine-tuning
+ The paper does not discuss the overheads and scalability of the planner, making it difficult to assess the usefulness of the proposed method
+ The method seems very sensitive to the choice of hyperparameters (when to unfreeze planner parameters during training, how often to use next action prediction vs next token prediction objective, etc.). The details of how to identify these hyperparameters efficiently is not provided

---

> ### Author Response · Authors · 2024-11-18
>
> Thank you for your review. We reply to your concerns below:
>
> ### Weaknesses
>
> > **1. The need for the planner module, and its advantages are unclear. In fact, the authors do not include any results in the paper showing that the incorporation of the planning module does indeed improve generation quality over vanilla training and fine-tuning.**
>
> Cornille et al. (2024) already showed that the incorporation of the planning module improves performance over vanilla training in a controlled experiment via the “Fixed” baseline: It uses the same adapter parameters that are used to fine-tune the LM, but instead of using a planner always selects the same action embedding. For clarity, we now include that baseline in our paper as well (Table 1). It performs similarly to the Uniform baseline we originally reported, i.e., consistently worse than models with a planner.
>
> > **2. The paper does not discuss the overheads and scalability of the planner, making it difficult to assess the usefulness of the proposed method.**
>
> We now discuss overheads and scalability of the planner in appendix F.
> Our current implementation introduces significant additional latency during LM fine-tuning—training duration is  3.9× using the Cornille et al (2024) planner, and 7.2× for our improved planner which uses soft-selection and end-to-end training compared to the Fixed baseline.
> While this is a significant overhead,it is important to note that the planner is only invoked once per sentence, rather than for every token.
> Hence, it should be possible to significantly reduce this overhead by optimizing the implementation for speed, which we did not aim for in our implementation.
>
> With regard to scalability, the planner size and latency is independent from the LLM architecture.
>
> > **3. The method seems very sensitive to the choice of hyperparameters (when to unfreeze planner parameters during training, how often to use next action prediction vs next token prediction objective, etc.). The details of how to identify these hyperparameters efficiently is not provided.**
>
> As with all deep learning systems, finding optimal hyperparameters is indeed crucial to get the best possible performance. However, for this work our goal is rather to understand the impact of retaining the next-action-prediction ability of the planner, rather than finding the optimal point to unfreeze. Hence, we do not perform extensive hyperparameter search, but picked “halfway” as a natural point to unfreeze.
>
> ---
>
> ### Requested Changes
>
> > **1. Quantify the training and inference overheads of using the planner module, in terms of latency and memory requirements. It looks like the planner adds additional training and inference latency, since both the planner and the original LLM now need to be evaluated.**
>
> See 2. above.
>
> > **2. Describe how planner size and latency change with LLM architecture. Do planner size and latency increase as LLM size increases, or can the same planner module be used irrespective of the underlying LLM architecture?**
>
> See 2. above.
>
> > **3. Add details on how the optimal hyperparameters for training the planner can be found when training on a new task or LLM. In particular, how can users determine when to unfreeze planner parameters during training, and how often (and during which iterations) to use next action prediction vs next token prediction objective? Can this process be automated? How long does this process take?**
>
> See 3. above.
>
> > **4. Add a new row to Table 1 with results on vanilla training and fine-tuning without the use of planner. The improvements from the planner, along with its overheads= numbers, will help better assess the significance of the work.**
>
> See 1 above.
>
> > **5. The paper is not self-contained in its current form. Some important details, such as the planner model architecture, are only present in the references, making it difficult for readers unfamiliar with the field. It would be good to add all necessary details required to reproduce this work in the paper, either in the main section or in the appendix.**
>
> We have now included the planner model architecture in our main section. Moreover, we will make code to reproduce our experiments available upon acceptance, which will allow readers to reproduce the work.

---

> ### Comment · Reviewer_3NNJ · 2024-11-25
> **Response to rebuttal**
>
> Thanks to the authors for the responses and revisions. My concerns about the need for planner, scalability and missing details are addressed. I also appreciate the authors for committing to release code. However, one main concern remains: The 7.2x increase in training cost with end-to-end planner training is extremely large, and the gains are modest (sometimes <0.1 increase in perplexity). Additionally, not all generation metrics are improved and some even become worse with the addition of the planner. Even if the 7.2x increase is reduced to only a 2x increase with a highly optimized implementation (which itself seems very  optimistic to me), it is unclear if the overhead is worth the modest gains. Can the authors comment on the real-world usefulness of the proposed method, and why users should use this in spite of the overheads? Also, what is the authors' estimate of the overheads with an optimized implementation during both training and inference averaged across the datasets used in the paper?

---

> > ### Author Response · Authors · 2024-11-26
> > **Response to "Response to rebuttal"**
> >
> > Thank you for the follow-up.
> > > **Can the authors comment on the real-world usefulness of the proposed method, and why users should use this in spite of the overheads?**
> >
> > As we stated in the Limitations section in the original draft, our work is not ready for real-world applicability yet: validating the method on production-scale models and optimizing its efficiency remain key steps.
> > However, we consider our insights into how end-to-end training can improve language modeling for the planner proposed in Cornille et al (2024) valuable to the community.
> >
> > > **Also, what is the authors' estimate of the overheads with an optimized implementation during both training and inference averaged across the datasets used in the paper?**
> >
> >
> > With an optimized implementation, the planner's forward/backward step should be as fast as or faster than that of the LM, given that the planner has significantly fewer parameters, especially compared to larger LMs.
> > Let $X$ denote the relative duration of one planner step compared to one LM step; in an optimized setup, $X \leq 1$.
> >
> >
> > **During training**, the LM makes predictions for all tokens in its context window in parallel using causal attention. Assuming a finite context window size $N$, this means that some predictions are made based on less than $N$ context tokens. However, this allows a significant speedup by letting the LM re-use computations for multiple predictions.
> > The planner architecture used in both Cornille et al (2024) and our paper contextualizes tokens first within-sentence and second across-sentences.
> > In the current unoptimized implementation, both the within- and the across-sentence contextualization is recomputed every time. An optimized implementation could make sure to avoid duplicating within-sentence computations, and parallelize across-sentence computations in a similar way to the LM.
> > This would allow the planner to make predictions for all sentences in its context window in parallel, which should result in only a *(1+X) relative overhead.
> >
> > **During inference**, because the planner is invoked once per sentence and there are about 35 tokens per sentence on average in our dataset, an optimized implementation should result in only a *(1+X/35) overhead.
> > Hence, while training overhead might still be somewhat significant even for an optimized implementation, we expect inference overhead should be minimal.

---

### Review · Reviewer_nujn · 2024-11-04

**Summary Of Contributions:**

This paper provides an end-to-end training approach to the planner-guided text generation LMs. Instead of separating the training into two stages with the planner and next-word generation, the paper warps the two training stages into one. A few approaches have been applied to avoid unstable training and improve the performance of end-to-end training, including the scheduled sampling of the planner to the LM, the weighted average of the planner to the LM, and freezing the planner. The paper investigates the effectiveness of the proposed method on GPT-2 and OLMo models trained on the Wikipedia English dataset. Ablation studies on different combinations of training approaches have been conducted to investigate the optimal approach for end-to-end training.

**Audience:**

Yes

**Claims And Evidence:**

Yes

**Requested Changes:**

1. Provide rigorous definitions of the notations and clear explanations of Figures 1 and 2.
2. Discuss the unfreezing schedule and how it differs from two-stage training, and clarify the optimal training method.
3. Can you discuss this method and the chain-of-thought approach for LM generation?
4. This might be minor, but adding experiments on other datasets would be beneficial.

**Strengths And Weaknesses:**

**Strength:**
1. Novelty: the paper proposes an end-to-end approach for planner+LM training.
2. Numerical evaluation: the paper provides a series of numerical evaluations on different combinations of the approaches and a thorough investigation of these combinations. The results serve as clear guidance on how to train planner-guided LMs.
3. Clarity: the experiment settings and results are clearly stated.

**Weakness:**
1. The experiment results in Table 1 suggest that NAP pretraining is needed (comparing no NAP pretraining and other). Also, unfreezing the planner halfway is better than unfreezing it immediately. Do these observations imply that the planner-guided LM requires different training stages (NAP pretraining -> LM fine-tuning -> planner + LM fine-tuning)? In such a case, calling it an end-to-end training might not be accurate.
2. The numerical evaluations are restricted to only one dataset. Adding results on more extensive datasets could be beneficial.
3. Figure 1 is a bit misleading. The action adapter only appeared in Figure 1. To my understanding, it should be a new component in the LM, as illustrated in Figure 2. However, the names are not consistent.
4. The notions are not well-defined. For examples:
    1. it is unclear how $\mathbf{z}_j$ is obtained in section 3.1
    2. $W_A$ in Figure 2 is not defined
    3. Eq(1) and (2) are not clearly stated (how they are used, they are not equations, what is one-hot encoding)
    4. what is the difference between $\mathbf{r}^l_j$ and $r^l_j$ in Eq(3) and section 3.2
    5. is $E^l_A(\cdot)$ a function?
    6. how does Figure 2 illustrate the evaluation procedure? Where $W_O, W_Q, W_K, W_V$ are introduced?
5.

---

> ### Author Response · Authors · 2024-11-18
>
> Thank you for your review. We reply to your concerns below:
>
>
> ### Weaknesses
>
>
> > **1. The experiment results in Table 1 suggest that NAP pretraining is needed (comparing no NAP pretraining and other). Also, unfreezing the planner halfway is better than unfreezing it immediately. Do these observations imply that the planner-guided LM requires different training stages (NAP pretraining -> LM fine-tuning -> planner + LM fine-tuning)? In such a case, calling it an end-to-end training might not be accurate.**
>
>
> Indeed, our experiments suggest the value of having an NAP pretraining stage. The reason we call it end-to-end is to emphasize that, unlike before, there is a stage where the planner parameters are optimized directly for the same objective as the LM (Next Token Prediction).
>
>
> > **2. The numerical evaluations are restricted to only one dataset. Adding results on more extensive datasets could be beneficial.**
>
>
> We now include results in Appendix D for an additional dataset (OpenWebText) comparing our best model, the best model of Cornille et al. (2024), and the baseline without a planner. OpenWebText covers a wider range of topics than Wikipedia. We observe that our best model consistently improves perplexity over the fixed baseline and over Cornille et al. (2024), though the improvement is smaller than for Wikipedia. We hypothesize this is because text covering a wider range of topics requires a more powerful planner with a larger action space trained on significantly more data. However, large-scale training is prohibitively expensive.
>
>
> > **3. Figure 1 is a bit misleading. The action adapter only appeared in Figure 1. To my understanding, it should be a new component in the LM, as illustrated in Figure 2. However, the names are not consistent.**
>
>
> Figure 2 shows the details of the action adapter in one layer, but the total adapter has parameters for multiple layers as we explain in Section 3.2. Figure 1 then makes an abstraction of these details. Because the action adapter parameters are different from the LM parameters, we think it is correct to show it as a separate component from the LM as we do now.
> We have clarified the connection between Figures 1 and 2 in the revised PDF.
>
>
> > **4. The notions are not well-defined. For examples:**
> > - **It is unclear how $z_j$ is obtained in Section 3.1.**
>
>
> $z_j$ is the output of pretrained SentenceBERT with input $t_j$. To clarify this, we now rephrase:
> *"To obtain training data for this planner, each sentence $t_j$ in the corpus is first embedded into a low-dimensional vector $\mathbf{z}_j$ using some pretrained text encoder (e.g., Sentence-BERT)."*
> to:
> *"To obtain training data for this planner, we apply a pretrained text encoder (e.g., Sentence-BERT) to each sentence $t_j$ in the corpus to produce a corresponding low-dimensional vector $\mathbf{z}_j$."*
>
>
> > - **WA in Figure 2 is not defined.**
>
>
> We now define $W_A$ in the caption of Figure 2.
>
>
> > - **Eq(1) and (2) are not clearly stated (how they are used, they are not equations, what is one-hot encoding).**
>
>
> We now clarify how Equations 1 and 2 are used, write them as equations, and explain one-hot encoding.
>
>
> > - **What is the difference between $r_{j}^l$ in Eq(3) and $r_{j}^l$ in Section 3.2?**
>
>
> $E_A$ is an embedding matrix as we state in Section 3.2. $E_A(\cdot)$ is a function that selects a column in the embedding matrix. $r_{j}^l$ in Section 3.2 is a column in the embedding matrix whose index is given by the $j$-th predicted action $\hat{a_j}$, while $r_{j}^l$ in Equation 3 is a weighted average of columns in the embedding matrix with weights given by $\text{softmax}(\mathbf{s})$. We now give explicit formulas for these, showing their differences.
>
>
> > - **Is $E_A(\cdot)$ a function?**
>
>
> Yes, $E_A(\cdot)$ is a function that selects a column in the embedding matrix.
>
>
> > - **How does Figure 2 illustrate the evaluation procedure?**
>
>
> Figure 2 is not intended to illustrate the evaluation procedure, but to illustrate the probing locations. We now clarify this in the revised PDF.
>
>
> > - **Where $W_O, W_Q, W_K, W_V$ are introduced?**
>
>
> We now explain $W_O, W_Q, W_K, W_V$ in the caption of Figure 2. They are the output, query, key, and value projection matrices of self-attention.
>
>
> ---
> (Requested Changes in below comment)

---

> ### Author Response · Authors · 2024-11-18
>
> (continued from above)
>
> ### Requested Changes
>
>
> > **1. Provide rigorous definitions of the notations and clear explanations of Figures 1 and 2.**
>
>
> See above.
>
>
> > **2. Discuss the unfreezing schedule and how it differs from two-stage training, and clarify the optimal training method.**
>
>
> Unfreezing the planner halfway through a single epoch with $N$ data points is indeed analogous to training for two epochs on $N/2$ data points and unfreezing after the first epoch, though not equivalent.
> We opt for the former approach because our constraint is computational budget, not data availability:
> Given a fixed computational budget, training for one epoch on all $N$ data points allows the model to experience the entire data distribution, which can be more beneficial than training for additional epochs on a smaller subset of the data.
> We now discuss this in section 3.3 of the revised PDF.
>
>
> > **3. Can you discuss this method and the chain-of-thought approach for LM generation?**
>
>
> Chain-of-thought reasoning improves outputs through explicit, token-level intermediate steps, and is not learned explicitly but only as a byproduct of LM pretraining.
> In contrast, the approach we build on conditions outputs on compact, latent-space plans, where a single plan embedding covers a sentence rather than individual tokens, and explicitly trains the model to condition generation on these plans. We have added this discussion to our Related Work in the revised PDF.
>
>
> > **4. This might be minor, but adding experiments on other datasets would be beneficial.**
>
>
> See point 2 above.

---

### Review · Reviewer_vyWt · 2024-11-05

**Summary Of Contributions:**

The paper proposes end-to-end training of the planner module with the language model, using a soft-max embedding of actions to ensure differentiable loss for the planner. It explores different training strategies, such as unfreezing the planner at various stages and combining losses during fine-tuning, along with a probing analysis of hidden representations across configurations.

**Audience:**

Yes

**Claims And Evidence:**

Yes

**Requested Changes:**

- Improve the clarity and organization of the experimental section.
- Provide standard deviation analysis to demonstrate statistical significance, and ideally include additional experiments on other tasks to strengthen the findings.
- Address the variability in impact that a single training configuration can have on different generation metrics. How could the authors unify or explain these differences across metrics?

**Strengths And Weaknesses:**

### Strengths
- This paper examines different methods for training improved planners for LLM generation, an important and timely topic.
- A variety of techniques are explored.

### Weaknesses and Questions
1. The primary concern the reviewer has is the significance of the results. A 0.09 difference in perplexity with OLMo is quite small; variations such as a different random seed or batching order could potentially alter it. Could the authors provide standard deviations for the results? Given the small differences and the varying relative order of the generation metrics, the results are less convincing.
2. The reviewer is also concerned about the choice of dataset. Currently, only Wikipedia data is used. Would a dataset covering a wider range of topics, such as for summarization or generation tasks, provide a more suitable setting? Extracting topics and styles with actions might also make results more interpretable.
3. The experimental section is challenging to follow. The reviewer found that it took long time to fully parse the authors’ description and what was really implemented.
4. When the authors mention using hard max for forward pass and soft max for backward pass, does this mean they evaluate with hard max but train with soft max? An unbiased evaluation would involve using a softmax average for both training and inference (to check if it performs well when evaluated as trained) or using random actions according to the softmax probabilities.
5. Important details are missing. Which optimizer is used? Is a fixed learning rate used without decay? Were all planner pretraining and finetuning runs done for only one epoch?
6. The statement "We explore this trade-off more in-depth by making models that mix oracle actions and predicted actions during pretraining" is confusing. From the reviewer’s understanding, there is no pretraining involved for the language model. When discussing oracle vs. predicted actions, the focus should be on LM finetuning (and planner finetuning), not pretraining.
7. How does the author mix oracle action and predicted action in Table 1? Which method is used?
8. The statement "the soft-selection *unfrz never* result shows that this alone is already beneficial, even without updating the planner" lacks a baseline for comparison. Why is there no straight-through result with *unfrz never*? This approach should be compared to baseline training but with a mix of OA and PA.
9. Although soft-selection improves perplexity, this is not consistently reflected in MAUVE or Edit metrics.
10. Terms such as "soft-mixing" and "softmix" are used interchangeably. Consistent terminology would improve readability.
11. Could the authors clarify what is meant by "distance to target token" in Figure 3a?
12. The colors in Figure 4 are difficult to distinguish.
13. In Figure 4, could the authors examine specific metrics for two groups — one with correctly predicted actions and one with incorrect predictions at inference? It could give a deeper understanding on how the planner output affect the performance.
14. Given the different curves of MAUVE, ROUGE-2, and Edit metrics, why do the authors conclude that “generation metrics benefit from having a model rely more on the planner output, even if it is imperfect”?

---

> ### Author Response · Authors · 2024-11-18
>
> Thank you for the detailed review. We reply to your concerns below:
>
> ### Weaknesses and Questions
>
> > **1. The primary concern the reviewer has is the significance of the results. A 0.09 difference in perplexity with OLMo is quite small; variations such as a different random seed or batching order could potentially alter it. Could the authors provide standard deviations for the results? Given the small differences and the varying relative order of the generation metrics, the results are less convincing.**
>
> We now show results for three repeats with different random seeds in Appendix C, demonstrating that we consistently improve perplexity for both GPT-2 and OLMo. We do not claim to improve performance in generation metrics: we discuss the trade-off between perplexity and generation metrics in Section 5.3.
>
> > **2. The reviewer is also concerned about the choice of dataset. Currently, only Wikipedia data is used. Would a dataset covering a wider range of topics, such as for summarization or generation tasks, provide a more suitable setting? Extracting topics and styles with actions might also make results more interpretable.**
>
> We now include results in Appendix D for an additional dataset (OpenWebText) comparing our best model, the best model of Cornille et al. (2024), and the baseline without a planner. OpenWebText covers a wider range of topics than Wikipedia. We observe that our best model consistently improves perplexity over the fixed baseline and over Cornille et al. (2024), though the improvement is smaller than for Wikipedia. We hypothesize this is because text covering a wider range of topics requires a more powerful planner with a larger action space trained on significantly more data. However, large-scale training is prohibitively expensive.
>
> > **3. The experimental section is challenging to follow. The reviewer found that it took a long time to fully parse the authors’ description and what was really implemented.**
>
> With “experimental section,” do you mean Section 4 (Experimental Setup) or Section 5 (Results and Discussion)? If you could detail which part was hard to follow, we will improve its clarity accordingly.
>
> > **4. When the authors mention using hard max for forward pass and soft max for backward pass, does this mean they evaluate with hard max but train with soft max? An unbiased evaluation would involve using a softmax average for both training and inference (to check if it performs well when evaluated as trained) or using random actions according to the softmax probabilities.**
>
> Your comment refers to our description of the straight-through estimator. Here, forward pass and backward pass do not refer to training and evaluation. Both refer to training: during the forward pass of training, the straight-through estimator uses a hard max. During the backward pass (i.e., during backpropagation), the gradient of the hard max is approximated  based on the assumption that a soft max was used in the forward pass. The evaluation using softmax for both training and inference you propose is exactly the model we propose, as shown in Equation (3).
>
> > **5. Important details are missing. Which optimizer is used? Is a fixed learning rate used without decay? Were all planner pretraining and finetuning runs done for only one epoch?**
>
> We use the Adam optimizer, with the base learning rate specified in Table 3 in Appendix A. Each run is indeed done for one epoch, as we are not constrained by data size. We now explicitly state the optimizer and epoch count in Appendix A.
>
> > **6. The statement "We explore this trade-off more in-depth by making models that mix oracle actions and predicted actions during pretraining" is confusing. From the reviewer’s understanding, there is no pretraining involved for the language model. When discussing oracle vs. predicted actions, the focus should be on LM finetuning (and planner finetuning), not pretraining.**
>
> Thank you for noticing this. We meant during LM finetuning rather than pretraining, and we have now corrected this.
>
> > **7. How does the author mix oracle action and predicted action in Table 1? Which method is used?**
>
> In Table 1, we do not show experiments that mix oracle and predicted actions. We show those experiments only in Figure 4. In Table 1, Cornille et al. (2024) OA is trained with 100% oracle actions, and all other runs are trained with 100% predicted actions.
>
> > **8. Why is there no straight-through result with unfrz never? This approach should be compared to baseline training but with a mix of OA and PA.**
>
> Straight-through with unfreeze-never is exactly equivalent to Cornille et al. (2024) PA, so we do compare to the relevant baseline. Straight-through acts identical to Cornille et al. (2024) PA in the forward pass, but as a softmax in backpropagating the gradient. However, if the gradient is never backpropagated into the planner (which is the case with unfreeze-never), it is in fact completely identical to Cornille et al. (2024).
>
> (Continued below)

---

> > ### Author Response · Authors · 2024-11-18
> >
> > (continued from above)
> >
> >
> > > **9. Although soft-selection improves perplexity, this is not consistently reflected in MAUVE or Edit metrics.**
> >
> > Indeed, perplexity improvement does not consistently translate into MAUVE or Edit distance improvements, and we do not claim that it does. We still consider a consistent improvement in perplexity valuable. We provide some analysis of the trade-off between perplexity and other metrics in Section 5.3.
> >
> > > **10. Terms such as "soft-mixing" and "softmix" are used interchangeably. Consistent terminology would improve readability.**
> >
> > We now consistently use "soft-selection" instead of "soft-mix" or "softmix" in the revised PDF.
> >
> > > **11. Could the authors clarify what is meant by "distance to target token" in Figure 3a?**
> >
> > The distance to the target token indicates how many positions into the future the probe’s target token is relative to the token immediately following the Language Model input. We now specify this in the revised PDF.
> > For example, if the text is “Thank you. No problem!” split into “Thank,” “you,” “.”, “No,” “problem,” and “!”, and we are at the point where the input is “Thank,” “you,” and “.”, and the target is “No,” the distance to the target token for “No” is 0, for “problem” is 1, for “!” is 2, and so on.
> >
> > > **12. The colors in Figure 4 are difficult to distinguish.**
> >
> > We have remade the graph with more distinct colors in the revised PDF, as well as different markers for the different lines.
> >
> > > **13. In Figure 4, could the authors examine specific metrics for two groups — one with correctly predicted actions and one with incorrect predictions at inference?**
> >
> > We now report a variant of Figure 4 that splits the metrics into groups with correct and incorrect actions in Appendix E in the updated PDF. We find that the group with correct actions is significantly better than the one with incorrect actions, and that this gap tends to become smaller for models trained on a bigger fraction of predicted actions. This makes sense, as predicted actions are less reliable, so the model becomes less sensitive to whether they are correct.
> >
> > Note that for generation metrics without any context (MAUVE and Latent Perplexity), there is no such thing as the correct action. For generation metrics with context, only the first text unit that is generated has a correct action associated with it. Later text units have generated tokens in their context, so there is no meaningful "true" correct action anymore.
> >
> > > **14. Given the different curves of MAUVE, ROUGE-2, and Edit metrics, why do the authors conclude that “generation metrics benefit from having a model rely more on the planner output, even if it is imperfect”?**
> >
> > Indeed only some generation metrics (Latent Perplexity and ROUGE-2) see a strong positive correlation with plan-matching accuracy. We now nuance our claim to reflect this.
> >
> > ---
> >
> > ### Requested Changes
> >
> > > 1. **Improve the clarity and organization of the experimental section.**
> >
> >    See point 4 above.
> >
> > > 2. **Provide standard deviation analysis to demonstrate statistical significance, and ideally include additional experiments on other tasks to strengthen the findings.**
> >
> >    See point 1 above.
> >
> > > 3. **Address the variability in impact that a single training configuration can have on different generation metrics. How could the authors unify or explain these differences across metrics?**
> >
> >    We now address the variance of generation metrics across random repeats in Appendix C.

---

### Decision · Action_Editor_vKca · 2024-12-18

**Recommendation:** Reject

**Comment:**

This paper builds on a technique for language modeling that is augmented via planning, similar to a form of hierarchical language modeling. The authors study a previously introduced technique where a pretrained planner module is used to predict actions that are then conditioned on during token generation. The authors show how to instead do this end-to-end.

The overall themes for this paper are interesting; a general study of when end-to-end techniques in language modeling are superior to the (often popular) methods that integrate pretrained components would be valuable. However, this paper felt a little bit limited, as it builds on one technique and comes with some potential downsides.

The reviewers were largely aligned around a set of concerns,
- Evidence for how valuable the end-to-end training was not particularly strong,
- Similar to the comment above, there are some questions around efficiency, and in particular, how valuable this technique could be.
The second point isn’t a killer all on its own, but it does suggest that the authors should at least discuss or even speculate on extensions of their results.

The first point could be alleviated with some more experiments that present stronger evidence of how much improvements end-to-end training can buy (perhaps even in the related areas the authors identified, i.e, other forms of hierarchical generation).

As a result the paper needs some more work, but certainly contains the start of a potentially strong work.

**Audience:**

Yes, although interest in this paper would be broadened if it had a wider focused.

**Claims And Evidence:**

As described in the full comments, more evidence would be useful.

**Resubmission Of Major Revision:**

The authors may consider submitting a major revision at a later time.